# Neuronal ensemble-specific DNA methylation strengthens engram stability

Kubra Gulmez Karaca [1,2], Janina Kupke[1], David V.C. Brito[1], Benjamin Zeuch[1], Christian Thome [3], Dieter Weichenhan[4], Pavlo Lutsik[4], Christoph Plass[4] & Ana M.M. Oliveira [1]*

Memories are encoded by memory traces or engrams, represented within subsets of neurons that are synchronously activated during learning. However, the molecular mechanisms that drive engram stabilization during consolidation and consequently ensure its reactivation by memory recall are not fully understood. In this study we manipulate, during memory consolidation, the levels of the de novo DNA methyltransferase 3a2 (Dnmt3a2) selectively within dentate gyrus neurons activated by fear conditioning. We found that Dnmt3a2 upregulation enhances memory performance in mice and improves the fidelity of reconstitution of the original neuronal ensemble upon memory retrieval. Moreover, similar manipulation in a sparse, non-engram subset of neurons does not bias engram allocation or modulate memory strength. We further show that neuronal Dnmt3a2 overexpression changes the DNA methylation profile of synaptic plasticity-related genes. Our data implicates DNA methylation selectively within neuronal ensembles as a mechanism of stabilizing engrams during consolidation that supports successful memory retrieval.

[1] Department of Neurobiology, Interdisciplinary Centre for Neurosciences (IZN), Heidelberg University, INF 366, 69120 Heidelberg, Germany. [2] Department of Cognitive Neuroscience, Donders Institute for Brain, Cognition and Behaviour, Radboud University Medical Center, Kapittelweg 29, 6525 EN Nijmegen, The Netherlands. [3] Department of Neurophysiology, Institute of Physiology and Pathophysiology, Heidelberg University, INF 326, 69120 Heidelberg, Germany. [4] Division of Cancer Epigenomics, German Cancer Research Center (DKFZ), INF 280, 69120 Heidelberg, Germany. *email: oliveira@nbio.uni-heidelberg.de

Neuronal ensembles, subset of neurons that exhibit coordinated activity during learning, are thought to hold unique memory representations or engrams. This idea was first supported by correlational studies showing that the neuronal ensembles activated during learning are reactivated by the retrieval of memory[1]. Recently, it was demonstrated that the reactivation of these neurons is necessary and sufficient to evoke memory recall[2–4]. Furthermore, a few studies have suggested mechanisms that determine which neurons will be recruited or allocated to the memory engram[5–7]. Nonetheless, we have a surprisingly limited understanding of the molecular mechanisms within behaviorally-allocated neurons that ensure the stabilization of the engram during memory consolidation required for successful reactivation at memory retrieval, which drives efficient memory recall.

Epigenetic mechanisms such as DNA methylation have emerged as important regulators of memory consolidation[8]. Rapid and dynamic changes in DNA methylation have been shown to occur in response to neuronal activity. Additionally, a number of studies have demonstrated that DNA methylation writers are required for memory formation[9]. However, these studies failed to address the contribution of DNA methylation-dependent mechanisms within neuronal ensembles to the formation and stabilization of the engram due to a lack of tools that allow for molecular manipulations specifically in neurons activated by learning.

Here, we hypothesized that DNA methylation within neuronal ensembles is a key determinant of engram stability in that it sets the likelihood of neuronal ensemble reactivation during retrieval and dictates the efficacy of memory recall. Recently developed technologies[10,11] allowed us to manipulate the levels of a DNA methyltransferase (Dnmt) specifically within hippocampal neuronal ensembles formed after contextual fear learning and to investigate the consequences of this manipulation for the strength of fear memory and stability of the engram. In particular, we elevated the levels of the de novo DNA methyltransferase (Dnmt3a2) in the dentate gyrus (DG) of the mouse hippocampus, as Dnmt3a2 expression is regulated by neuronal activity[12] and has established functions in long-lasting neuronal adaptations[13–15]. We showed that reinforcing DNA methylation-related mechanisms through Dnmt3a2 overexpression within memory-encoding neuronal ensembles specifically during consolidation was sufficient to strengthen contextual fear memory and the stability of the engram. Specifically, we found that Dnmt3a2 overexpression within the hippocampal engram improved the fidelity of its reactivation upon memory retrieval. We further demonstrated that reinforcing DNA methylation-related mechanisms in a sparse, non-engram subset of neurons prior to learning or during memory consolidation did not bias engram allocation or modulate memory strength, respectively. Moreover, we found that Dnmt3a2 overexpression in cultured hippocampal neurons modified the DNA methylation landscape of several genes with critical roles in synaptic plasticity and learning and memory.

In summary, here we identify an epigenetic factor in neuronal ensembles that primes the behaviorally-allocated neuronal ensemble for improved fidelity in cellular reactivation at the time of memory recall, resulting in enhanced behavioral response. Our findings show sufficiency for epigenetic regulation in hippocampal memory consolidation at the level of neuronal ensembles. While we focus here on hippocampal engrams, we suggest that epigenetic mechanisms are likely to be preserved across different regions of the central nervous system and during diverse adaptive processes.

## Results

### Dnmt3a2 overexpression in dentate gyrus neuronal ensembles.
We aimed at manipulating the molecular composition of neurons within behaviorally-allocated neuronal ensembles to investigate

whether DNA methylation within these cells serves as a mechanism regulating engram stability and long-term memory storage. To this end, we generated recombinant adeno-associated viruses (rAAVs) that allow the expression of a de novo DNA methyltransferase, Dnmt3a2, or a control protein (GFP or LacZ) in an activity-dependent manner. To achieve the neuronal activity-induced expression of these exogenous genes, we used the synthetic neuronal activity-dependent promoter E-SARE (enhanced synaptic activity-regulated element)[16] (Fig. 1a). This promoter has been previously shown to be rapidly activated by neuronal activity and to reliably label those neurons that were activated by experience[16,17]. Therefore, it is a suitable tool to achieve the rapid and targeted expression of exogenous genes in neuronal ensembles. We first confirmed that the expression of E-SARE-driven proteins (HA-tagged GFP (HA-GFP) or Dnmt3a2 (HA-Dnmt3a2)) was strictly regulated by neuronal activity and that high expression levels are achieved at early time points after the induction of neuronal activity (Supplementary Fig. 1a–c). In order to determine the permanence of the exogenous proteins (HA-GFP and HA-Dnmt3a2) in the neurons, we performed a longer time course analysis (Supplementary Fig. 1d, e). Neuronal activity was blocked 4 h after bicuculline treatment with the sodium channel blocker tetrodotoxin (TTX) to prevent the synthesis of new transcripts. We found that 48–72 h after the induction of neuronal activity, the levels of HA-GFP and HA-Dnmt3a2 were significantly reduced and comparable to baseline. We further confirmed that HA-Dnmt3a2 is catalytically active. We found that after induction of synaptic activity, levels of HA-Dnmt3a2 expression and DNA methylation per cell were positively correlated in hippocampal cultures (Fig. 1b).

Next, we stereotaxically delivered the rAAVs selectively into the mouse DG. Similar to the experiments in culture, E-SARE-driven expression was tightly regulated by neuronal activity and detectable 2 h after kainic acid administration or exposure to a novel context (Supplementary Fig. 2a, b). This shows that the kinetics of E-SARE-driven expression is comparable to that of endogenous immediate early genes (IEGs). It is important to note that similar expression kinetics would not be achievable with tools based on the Tet-OFF system[18,19]. Furthermore, the analysis of endogenous Arc, which labels behaviorally allocated neuronal ensembles, and E-SARE-driven expression in the DG after novel environment exposure revealed that the colocalization of Arc+ and GFP+ neuronal populations is significantly higher than expected by chance (Supplementary Fig. 2c–e). Moreover, we observed that at least 75% of Arc-expressing neurons also displayed GFP signal (Supplementary Fig. 2d). Accordingly, we concluded that this approach can be used to reliably induce the expression of a DNA methylation writer within more than 75% of behaviorally-allocated DG neurons during the early stages of memory consolidation.

### Neuronal ensemble Dnmt3a2 overexpression enhances memory.
Next, we determined the effect of Dnmt3a2 overexpression within DG ensembles on cognitive function. Before performing the memory test, we habituated the mice to the experimental room and experimenter at a time point for which viral expression was confirmed to be almost undetectable[14,20] (Fig. 1c). This was done to minimize Dnmt3a2 overexpression prior to the learning phase. We confirmed that contextual fear conditioning (CFC) training, similarly to exposure to a novel context, also leads to the activation of E-SARE-driven expression at early time points after learning (2 h, Supplementary Fig. 2f, g). On the experiment day, we trained both experimental groups (Control or E-SARE-HA-Dnmt3a2) in CFC, which leads to the expression of a control gene (HA-GFP or HA-LacZ) or HA-Dnmt3a2 in behaviorally allocated

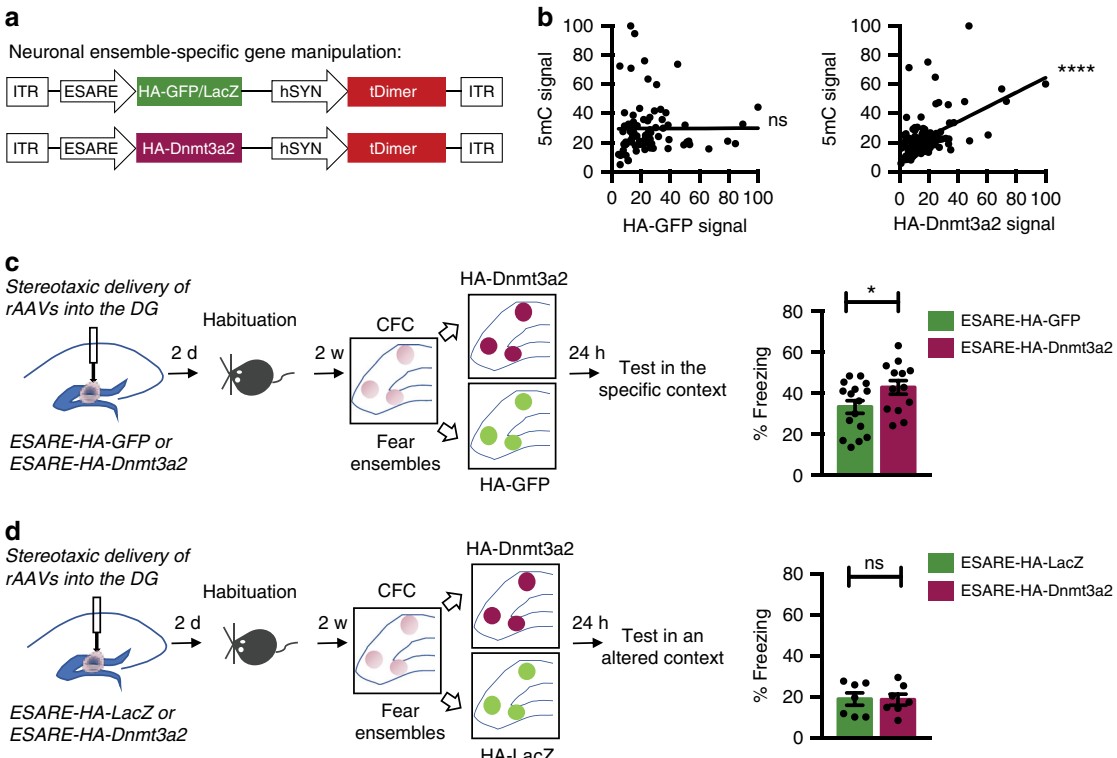

**Fig. 1 Neuronal ensemble-specific Dnmt3a2 overexpression enhances fear memory. a** Schematic representation of the viral constructs used for neuronal ensemble targeted gene manipulation. E-SARE promoter drives the expression of HA-GFP, HA-LacZ or HA-Dnmt3a2. Constitutive tDimer expression (human Synapsin promoter) serves as an infection marker. **b** Correlation analysis of 5mC and HA immunocytochemical signal intensities in primary hippocampal cultures. Cultures were infected with E-SARE-HA-GFP ($n = 4$ independent cell preparations, Pearson's $r = 0.0035$, $p = 0.9756$) or E-SARE-HA-Dnmt3a2 ($n = 4$ independent cell preparations, Pearson's $r = 0.5174$, $p < 0.0001$). **c** Experimental schedule used to assess the effect of Dnmt3a2 overexpression in the neuronal ensembles in contextual fear memory. Graph represents % time freezing in the conditioned context. Mice were injected with E-SARE-HA-GFP or E-SARE-HA-Dnmt3a2 viruses (E-SARE-HA-GFP ($n = 16$) vs E-SARE-HA-Dnmt3a2 ($n = 13$), t(27) = 2.083, $p = 0.0468$ by unpaired t-test). **d** Experimental schedule used to assess the effect of Dnmt3a2 overexpression in the neuronal ensembles in freezing in an altered context. Graph represents % time mice spent freezing in the altered context. Mice were injected with E-SARE-HA-GFP or E-SARE-HA-Dnmt3a2 viruses (E-SARE-HA-LacZ ($n = 7$) vs E-SARE-HA-Dnmt3a2 ($n = 7$), t(12) = 0.06332, $p = 0.9506$ by unpaired t-test). ITR: inverted terminal repeat, w- week, DG: Dentate gyrus of the hippocampus, CFC: contextual fear conditioning, rAAVs: recombinant adeno-associated viruses. *$p < 0.05$; ****$p < 0.0001$, ns: not significant by the respective statistical test. Error bars represent s.e.m. Source data are provided as a Source Data file.

DG ensembles. When the mice were tested in the conditioning context 24 h later, the E-SARE-HA-Dnmt3a2 group exhibited significantly higher freezing levels than the control group, indicating enhanced memory (Fig. 1c). We did not detect alterations in the mice's basal freezing rates or reaction to the shock during the training session that could account for the observed difference in freezing responses during the test (Supplementary Fig. 3a). Further, we showed that the E-SARE-HA-Dnmt3a2 group exhibited similar freezing rates to the control group when tested in an altered context 24 h after conditioning (Fig. 1d), verifying that the observed memory enhancement was context-specific and not due to a general and unspecific increased fear response. We confirmed that both groups displayed similar overall activities during the training session (Supplementary Fig. 3b).

Next, we investigated whether Dnmt3a2 overexpression must occur within behaviorally allocated neuronal ensembles in order to dictate cognitive abilities in mice. To this end, we used a previously characterized Cre/loxP recombination system[14] and took advantage of its intrinsic leakiness to achieve sparse expression of GFP or HA-Dnmt3a2 in a similarly-sized, but random subset of DG neurons (Fig. 2a). Using this approach, we could, in the absence of tamoxifen, achieve recombination in a sparse population of DG neurons (5.682 % ± 0.6181 (mean ± s.e.m.) DG neurons expressed HA-Dnmt3a2, $n = 4$) (Supplementary Fig. 4a) of similar size to

the ensemble labeled with E-SARE after CFC training (4.793 % ± 0.3543 DG neurons labeled using ESARE-HA-Dnmt3a2, $n = 4$) (Supplementary Fig. 2g). Moreover, we confirmed that similarly to the E-SARE-dependent system, HA-Dnmt3a2 expression using the Cre/loxP strategy led to DNA methylation changes in DG neurons (Supplementary Fig. 4b, c). We stereotaxically delivered Cre-loxP-GFP or Cre-loxP-HA-Dnmt3a2 rAAVs into the DG of the mice and carried the same behavioral schedule as in the ensemble-specific manipulation (Fig. 2b). We found that overexpression of Dnmt3a2 in a sparse and random DG neuronal population did not enhance fear memory (Fig. 2b). Notably, the overall activity and reaction to shock administration in both of the groups was similar during the training session (Supplementary Fig. 4d). These results clearly indicate that Dnmt3a2 overexpression must be confined to the behaviorally allocated neuronal ensemble in order to dictate memory strength. To the best of our knowledge, these findings demonstrate for the first time that molecular manipulation selectively in a defined neuronal population activated during learning is sufficient to modulate memory formation.

**Non-ensemble Dnmt3a2 overexpression does not allocate memory.** Next we aimed at investigating if DNA methylation mechanisms function selectively during memory consolidation or

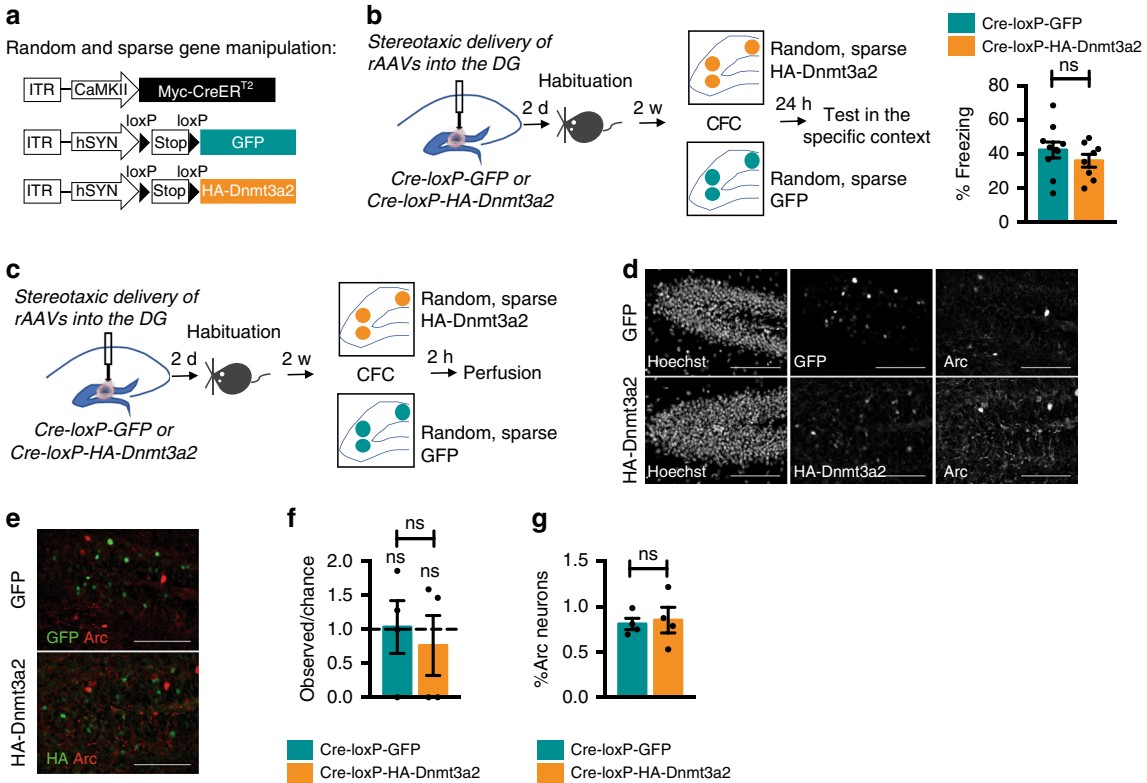

**Fig. 2 Sparse and random Dnmt3a2 overexpression does not allocate or enhance memory. a** Schematic representation of the viral constructs used to express GFP or HA-Dnmt3a2 in a random and sparse subset of DG neurons[14]. **b** Experimental schedule used to assess the effect of DG random and sparse Dnmt3a2 overexpression in contextual fear memory. Graph represents % time freezing in the conditioned context. Mice were injected with CaMKII-mycCreER[T2] and hSyn-loxP-GFP (Cre-loxP-GFP) or hSyn-loxP-HA-Dnmt3a2 (Cre-loxP-HA-Dnmt3a2) (Cre-loxP-GFP ($n = 10$) vs Cre-loxP-HA-Dnmt3a2 ($n = 8$), t(16) = 1.022, $p = 0.3222$ by unpaired $t$-test). **c** Experimental schedule used to assess the effect of DG random and sparse Dnmt3a2 overexpression in neuronal ensemble allocation. **d, e** Representative images of immunohistochemical analysis of GFP, HA-Dnmt3a2 and Arc expressions of brain slices of mice injected with Cre-loxP-GFP or Cre-loxP-HA-Dnmt3a2. Scale bar represents 50 μm. **f** Observed over chance overlap between GFP or HA-Dnmt3a2 and Arc expressions in the DG of mice injected with Cre-loxP-GFP or Cre-loxP-HA-Dnmt3a2 (Observed overlap vs chance overlap: GFP: $n = 4$, t(3) = 0.5203, $p = 0.6388$; HA-Dnmt3a2: $n = 4$, t(3) = 0.768, $p = 0.4984$ by paired $t$-test; Observed/chance GFP ($n = 4$) vs HA-Dnmt3a2 ($n = 4$), t(6) = 0.4584, $p = 0.6628$ by unpaired $t$-test). Dashed line represents the chance level. **g** Percentage of Arc positive neurons in the DG of mice injected with Cre-loxP-GFP or Cre-loxP-HA-Dnmt3a2 ($n = 4$, t(6) = 0.2721, $p = 0.7947$ by unpaired $t$-test). ITR: inverted terminal repeat, w- week, DG: dentate gyrus of the hippocampus, CFC: contextual fear conditioning, rAAVs: recombinant adeno-associated viruses. ns: not significant by the respective statistical test. Error bars represent s.e.m. Source data are provided as a Source Data file.

whether they also bias the recruitment of individual neurons to a memory engram. To this end, we stereotaxically delivered Cre-loxP-GFP or Cre-loxP-HA-Dnmt3a2 viral constructs into the mouse DG to achieve the random and sparse expression of GFP or HA-Dnmt3a2 already prior to learning (Fig. 2c). We performed CFC and sacrificed the mice 2 h later and, using endogenous Arc expression as a marker, characterized the neuronal population activated by learning (Fig. 2c). Colocalization analysis showed that neither GFP+ nor HA-Dnmt3a2+ neurons overlapped with Arc+ neurons above the chance level (Fig. 2d–f). Moreover, the ratio of overlap observed between Arc and HA-GFP or HA-Dnmt3a2 and the overlap that might occur by chance was similar in both Cre-loxP-HA-Dnmt3a2 and Cre-loxP-GFP groups and not significantly different from the chance level (Fig. 2f). Additionally, HA-Dnmt3a2 overexpression per se did not alter the ensemble size, as both groups displayed similar percentages of Arc+ neurons (Fig. 2g). Moreover, no differences were detected in overall activity or reaction to shock administration between the two groups (Supplementary Fig. 4e). Thus, this set of experiments showed that overexpression of Dnmt3a2 in a sparse DG neuronal population prior to learning does not bias their integration into a fear memory engram. These findings further suggest that the function of Dnmt3a2 in memory engram

modulation is restricted to behaviorally allocated neurons and to the memory consolidation phase.

**Increased Dnmt3a2 in engram biases reactivation at recall.** Next, we sought to identify the underlying mechanism for the context-specific cognitive enhancement achieved by Dnmt3a2 overexpression in neuronal ensembles. It is known that neuronal ensembles active during encoding are reactivated at memory recall at above chance levels[4,21]. We hypothesized that a Dnmt3a2-overexpressing neuronal ensemble would display increased stability, rendering it more likely to be reactivated during memory recall, and resulting in more efficient memory retrieval. To test this hypothesis, we assessed the reactivation of the learning-associated ensemble at recall. To label the neuronal ensembles formed by learning, we used the recently developed Robust Activity Marking (RAM) system[19] (Supplementary Fig. 5a). RAM is a modified Tet-OFF system with a strict temporal control and regulation by spatial learning that reliably marks neuronal ensembles in the DG (Supplementary Fig. 5b, c). Importantly, the reporter protein is only detectable from 16 h after a neuronal stimulus[19] and not at earlier time points (2 h, Supplementary Fig. 5d, e). Thus, for analyses performed within a

few hours after memory recall, use of the RAM systems allows for the exclusive detection of neurons activated during learning without the contamination by neurons activated during memory recall (note that our analysis was performed 2 h after recall). After confirming that E-SARE and RAM neuronal activity marking tools significantly label an overlapping population of neurons after a learning experience (Supplementary Fig. 6), we delivered a combination of rAAVs into the DG: E-SARE viral vectors were employed to achieve Dnmt3a2 or LacZ expression immediately after learning and pRAM-myc-GFP were used to label learning-induced neuronal ensembles. We then applied the same experimental schedule as before, with the exception that mice were kept on a doxycycline diet until 3 days prior to the behavioral experiment (Fig. 3a). We confirmed that also in this set up, both of the experimental groups displayed similar overall activity levels in the training session (Supplementary Fig. 7a). As expected, the overlap between the neuronal populations activated by learning (labeled with myc-GFP) and by memory recall (identified by Fos expression) was significantly higher than predicted by chance in both control and Dnmt3a2 overexpressing groups (Fig. 3b, c). This confirms that re-exposure to a previously visited context reactivates an overlapping neuronal population. Interestingly in the E-SARE-HA-Dnmt3a2 group the observed overlap above what would occur by chance was significantly higher than in the control group (Fig. 3b, c). Moreover, the E-SARE-HA-Dnmt3a2 group displayed an improved reactivation rate and exhibited a higher percentage of neurons expressing both GFP and Fos compared to control (Fig. 3d, e). A detailed analysis of the DG blades showed that these effects originated primarily from the upper blade (Fig. 3c–e). We then applied a second formula to assess engram stability, the similarity index, which measures the degree of the similarity in the pattern of activated cells between two events[22]. Similar to the reactivation rate, E-SARE-HA-Dnmt3a2 mice displayed significantly higher similarity indices between learning and memory recall ensembles specifically in the upper blade of the DG (Supplementary Fig. 7b). Importantly, the identified differences between the two groups were not due to differences in the size of the neuronal population activated by memory recall (Supplementary Fig. 7c).

Furthermore, we ruled out the possibility that neurons activated during recall were increasingly represented in the population with higher Dnmt3a2 levels due to a direct effect on the transcription of IEGs. To this end, we examined the likelihood that a random subset of neurons with higher Dnmt3a2 levels would express IEGs after fear memory recall, and found that this was not the case (Supplementary Fig. 7d–f). These observations demonstrate that Dnmt3a2 overexpression by learning-allocated neuronal ensembles generates a more stable engram characterized by an improved reactivation response to memory recall.

Finally, we tested if the improved reactivation and similarity indices were specific to retrieval of the memory encoded by the manipulated neuronal ensemble. To this end, we determined the degree of overlap between the DG upper blade neuronal population activated by fear conditioning and that activated by testing in a novel context (Fig. 4a). We found that neither the GFP- nor the Dnmt3a2-expressing group displayed an overlap between novel context- and learning-activated ensembles above chance level (Fig. 4b, c), confirming that the exposure to distinct contexts activates dissimilar neuronal populations. Moreover, we observed that the observed- to chance-overlap ratio was equivalent in the two groups (Fig. 4c), and that E-SARE-HA-Dnmt3a2 expressing ensembles in the upper blade of the DG did not show a preferential reactivation of the initial ensemble upon exposure to the novel context (Fig. 4d). Neither the degree of overlapping neurons (Fig. 4e), nor the similarity index (Supplementary Fig. 8a) in the upper blade of the DG of both groups

were different in an altered context. The proportion of neurons activated upon altered context exposure was also similar between the two groups (Supplementary Fig. 8b). Importantly, we confirmed that the training induced a similar proportion of myc-GFP expressing neurons in the two groups in all of the training sessions (Supplementary Fig. 8c). Overall, this set of experiments demonstrates that Dnmt3a2-overexpressing neuronal ensembles are more likely to be reactivated specifically by recall of the fearful context.

**Dnmt3a2 alters DNA methylation in memory-related genes.** In order to gain insight into the mechanisms associated with improved memory and neuronal ensemble reactivation in conditions of Dnmt3a2 overexpression, we identified DNA methylation changes triggered by HA-Dnmt3a2 expression. We performed whole-genome bisulfite sequencing (WGBS) of hippocampal cultures that were infected with E-SARE-HA-GFP or E-SARE-HA-Dnmt3a2 and treated 4 h with bicuculline to trigger the expression of HA-GFP or HA-Dnmt3a2 (Fig. 5a). The technical quality of the WGBS data were high, with bisulfite conversion rate ranging between 0.988 and 0.990, and average depth of coverage between 16.0 and 20.0 single-stranded. The comparison of the DNA methylation pattern of cells expressing HA-Dnmt3a2 or HA-GFP revealed 224 differentially methylated regions (DMRs) (Supplementary Data 1). As expected, the majority of the DMRs (199 or 89%) were hypermethylated in the HA-Dnmt3a2 overexpression condition (Fig. 5b and Supplementary Data 1). Analysis of the genomic distribution of the DMRs showed preferential location in intergenic (36%) and intronic (35%) regions (Fig. 5c). Despite the enrichment in intergenic regions, the DMRs mapped to the vicinity of 213 genes. Finally, we examined which functional categories were over represented among the DMR-associated genes. Remarkably, differential methylation preferentially targeted genes involved in synaptic plasticity, as evidenced by the enrichment of GO terms "NMDA receptor signaling pathway", "synapse organization", "organization of synapse structure or activity" (Fig. 5d and Supplementary Data 2). Representative examples are the genes Tiam1[23–26], Cabp1[27–29] and Lrrtm1[30,31] that have well-established roles in activity-dependent structural plasticity, synaptic strength and learning and memory (Supplementary Fig. 9b). Taken together, these findings suggest that Dnmt3a2 through epigenetic regulation of synaptic plasticity-related genes impacts memory formation and dictates the likelihood of neuronal ensemble reactivation (Fig. 6).

**Discussion**
In this study, we manipulated the molecular composition of hippocampal neuronal ensembles selectively during fear memory consolidation and identified an epigenetic mechanism that is sufficient to enhance memory engram stability. We showed that increasing the levels of a DNA methyltransferase, Dnmt3a2, within behaviorally-allocated neuronal ensembles reinforced engram maintenance during memory consolidation, that resulted in the improved fidelity of engram reactivation during memory recall and the strengthening of long-term memory storage (Fig. 6). Moreover, we showed that this manipulation leads to DNA methylation changes in genes with established functions in synaptic plasticity and memory formation.

Previous studies performed artificial activation or inhibition (using optogenetic or chemogenetic tools) of memory-encoding neuronal ensembles during memory retrieval and provided compelling evidence that this subset of neurons holds the representation of a particular learned experience[2,4,32–36]. These studies confirmed earlier correlational observations[1] that suggested that

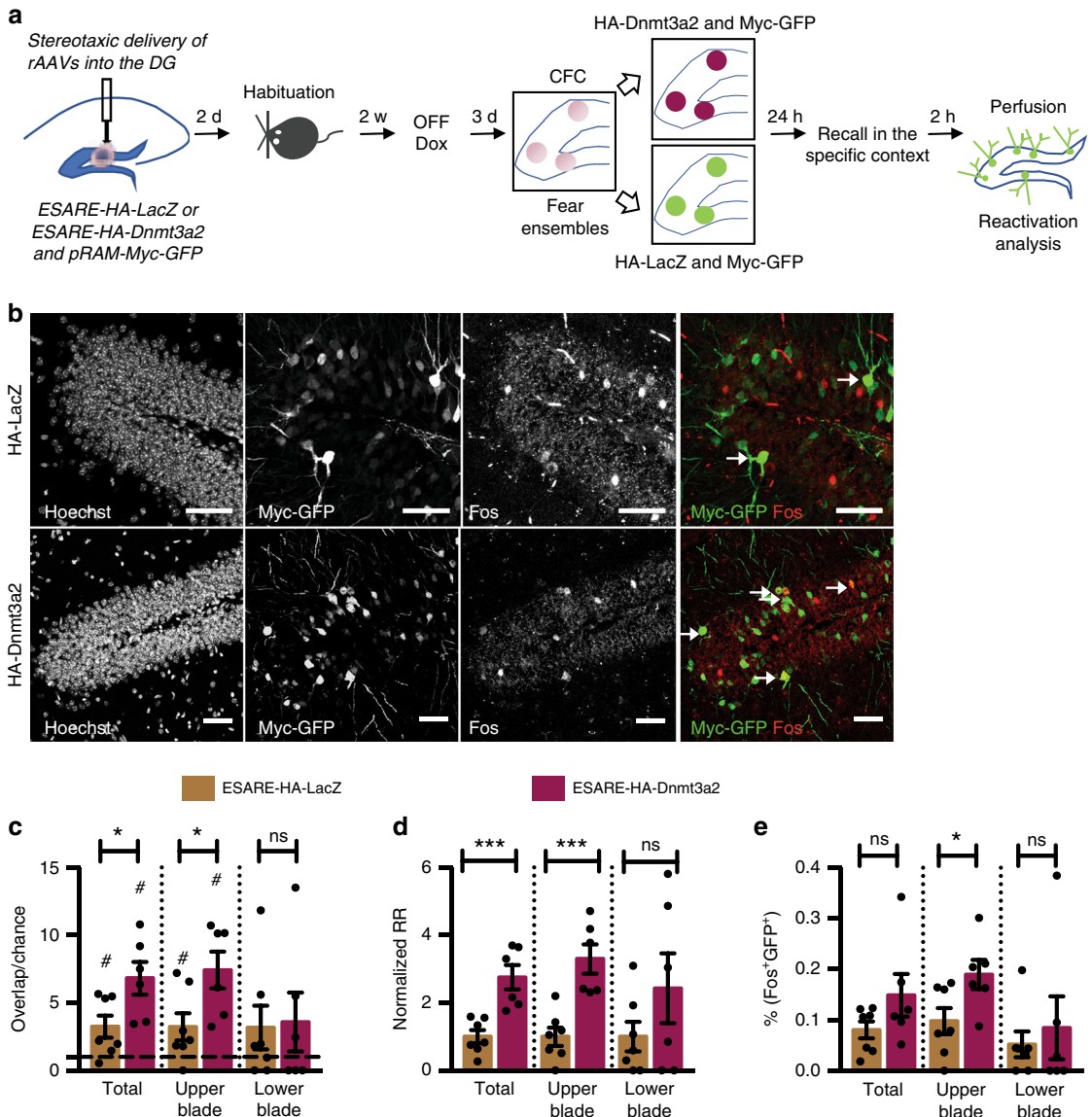

**Fig. 3 Ensemble-specific Dnmt3a2 overexpression facilitates recall-dependent reactivation. a** Experimental schedule used to assess the engram stability at memory recall. **b** Representative images of immunohistochemical analysis of myc-GFP and Fos in mice injected with RAM-myc-GFP and E-SARE-HA-LacZ or E-SARE-HA-Dnmt3a2. Scale bar represents 50 μm. **c** Observed over chance overlap between myc-GFP and Fos expressions in mice injected with RAM-myc-GFP and E-SARE-HA-LacZ or E-SARE-HA-Dnmt3a2 (Observed- vs chance-overlap: Total HA-LacZ: $n = 7$, $W = 26$, $p = 0.0312$ by Wilcoxon test; Total HA-Dnmt3a2: $n = 6$, $W = 21$, $p = 0.0312$ by Wilcoxon test; Upper blade HA-LacZ: $n = 7$, $W = 24$, $p = 0.0469$ by Wilcoxon test; Upper blade HA-Dnmt3a2: $n = 6$, $W = 21$, $p = 0.0312$ by Wilcoxon test; Lower blade HA-LacZ: $n = 7$, $W = 10$, $p = 0.4688$ by Wilcoxon test; Lower blade HA-Dnmt3a2: $n = 6$, $W = 9$, $p = 0.4375$ by Wilcoxon test) (Observed/chance: Total: HA-LacZ ($n = 7$) vs HA-Dnmt3a2 ($n = 6$), t(11) = 2.516, $p = 0.0287$ by unpaired $t$-test; Upper blade: HA-LacZ ($n = 7$) vs HA-Dnmt3a2 ($n = 6$), t(11) = 2.523, $p = 0.0283$ by unpaired $t$-test; Lower blade: HA-LacZ ($n = 7$) vs HA-Dnmt3a2 ($n = 6$), $U = 20$, $p = 0.9417$ by Mann–Whitney test). Dashed line represents the chance level. **d** Analysis of the reactivation of the learning-activated neuronal population during memory recall in mice injected with RAM-myc-GFP and E-SARE-HA-LacZ or E-SARE-HA-Dnmt3a2 (Total: HA-LacZ ($n = 7$) vs HA-Dnmt3a2 ($n = 6$), t(11) = 4.473, $p = 0.0009$ by unpaired $t$-test; Upper blade: HA-LacZ ($n = 7$) vs HA-Dnmt3a2 ($n = 6$), t(11) = 4.643 $p = 0.0007$ by unpaired $t$-test; Lower blade HA-LacZ ($n = 7$) vs HA-Dnmt3a2 ($n = 6$), t(11) = 1.346, $p = 0.2054$ by unpaired $t$-test). **e** Percentage of GFP$^+$ and Fos$^+$ neurons after memory recall in mice injected with RAM-myc-GFP and E-SARE-HA-LacZ or E-SARE-HA-Dnmt3a2 (Total: HA-LacZ ($n = 7$) vs HA-Dnmt3a2 ($n = 6$), t(11) = 1.602, $p = 0.1375$ by unpaired $t$-test; Upper blade: HA-LacZ ($n = 7$) vs HA-Dnmt3a2 ($n = 6$), t(11) = 2.378, $p = 0.0366$ by unpaired $t$-test; Lower blade: HA-LacZ ($n = 7$) vs HA-Dnmt3a2 ($n = 6$), $U = 18$, $p = 0.7203$ by Mann–Whitney test). RR: reactivation rate; w- week, DG: dentate gyrus of the hippocampus, CFC: contextual fear conditioning, Dox: doxycycline, rAAVs: recombinant adeno-associated viruses. *, $^{\#}p < 0.05$; ***$p < 0.001$; ns: not significant by the respective statistical test. Error bars represent s.e.m. Source data are provided as a Source Data file.

the spatially distributed subset of neurons activated during learning (neuronal ensembles) must be stabilized in order to be recruited as a group, and to successfully recreate aspects of a particular learning event. However, at a molecular level, what regulates and determines this stability has been unclear. Although

epigenetic mechanisms are known to regulate several plasticity-related processes[8], their role in the stabilization of cellular memory representations has never been addressed. Here, we interrogated the spatiotemporal sufficiency of DNA methylation-dependent mechanisms for the modulation of a fear memory engram.

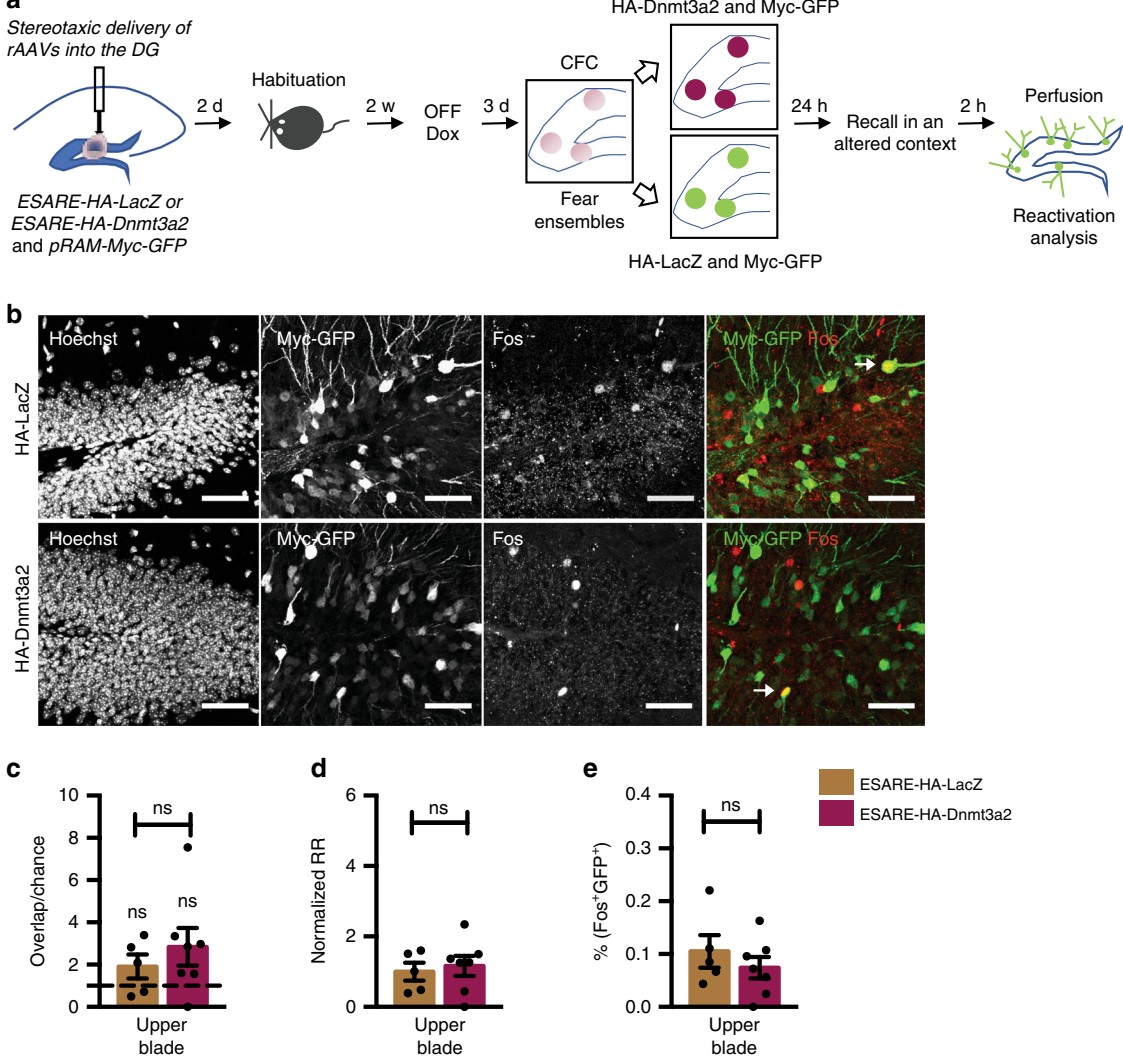

**Fig. 4 Dnmt3a2 overexpressing neuronal ensembles enhanced stability is context-specific. a** Experimental schedule used to assess the engram stability at memory recall in a novel context. **b** Representative images of immunohistochemical analysis of myc-GFP and Fos in mice injected with RAM-myc-GFP and E-SARE-HA-LacZ or E-SARE-HA-Dnmt3a2. Scale bar represents 50 μm. **c** Observed over chance overlap between myc-GFP and Fos expressions in mice injected with RAM-myc-GFP and E-SARE-HA-LacZ or E-SARE-HA-Dnmt3a2 (Observed overlap vs chance overlap: HA-LacZ: $n = 5$, t(4) = 2.605, $p = 0.0597$ by paired *t*-test; HA-Dnmt3a2: $n = 7$, W = 12, $p = 0.3750$ by Wilcoxon test; Observed/chance HA-LacZ ($n = 5$) vs HA-Dnmt3a2 ($n = 7$), t(10) = 0.797, $p = 0.4440$ by unpaired *t*-test). Dashed line represents the chance level. **d** Analysis of the reactivation of the learning-activated neuronal population during memory recall in an altered context in mice injected with RAM-myc-GFP and E-SARE-HA-LacZ or E-SARE-HA-Dnmt3a2 (HA-LacZ ($n = 5$) vs HA-Dnmt3a2 ($n = 7$), t(10) = 0.4158, $p = 0.6864$ by unpaired *t*-test). **e** Percentage of GFP and Fos positive neurons after memory recall in an altered context in mice injected with RAM-myc-GFP and E-SARE-HA-LacZ or E-SARE-HA-Dnmt3a2 (HA-LacZ ($n = 5$) vs HA-Dnmt3a2 ($n = 7$), t(10) = 0.8695, $p = 0.4050$ by unpaired *t*-test). RR: Reactivation rate; w- week, DG: dentate gyrus of the hippocampus, CFC: contextual fear conditioning, Dox: doxycycline, rAAVs: recombinant adeno-associated viruses. ns: not significant by the respective statistical test. Error bars represent s.e.m. Source data are provided as a Source Data file.

Pharmacological and genetic evidence emerging from different laboratories consistently showed that the inhibition of DNA methyltransferases function, and particularly that of Dnmt3a, impairs hippocampal long-term plasticity and memory formation[9]. The abolishment of Dnmt3a expression, but not that of Dnmt1 or Dnmt3b, in the rat hippocampus suggested a selective requirement for Dnmt3a in hippocampus-dependent memory[37]. Two other studies analyzed the memory performance of conditional knockout mice in which the *Dnmt3a* gene locus was disrupted[38,39]. These studies produced seemingly contrasting results; in one study *Dnmt3a* deletion in the mouse forebrain impaired long-term memory formation[39] whereas in the other did not impact learning and memory[38]. Both studies found that

Dnmt1 deletion does not impact long-term memory[38,39]. The reason for the discrepancy is not clear but may be attributed to differences in the gender, age or genetic background of the mice used. Our previous work is in line with a requirement for Dnmt3a function in memory formation. Taken together, these studies indicated that particularly the function of Dnmt3a is important for hippocampal memory formation. We showed that the Dnmt3a isoform, Dnmt3a2 is required for different forms of long-lasting neuronal adaptions, including long-term memory formation, and that the expression of *Dnmt3a2*, but not *Dnmt3a1*, is induced by neuronal activity and learning[12–15]. Therefore in the current study, we employed activity-dependent tools to overexpress and reinforce the function of the activity-

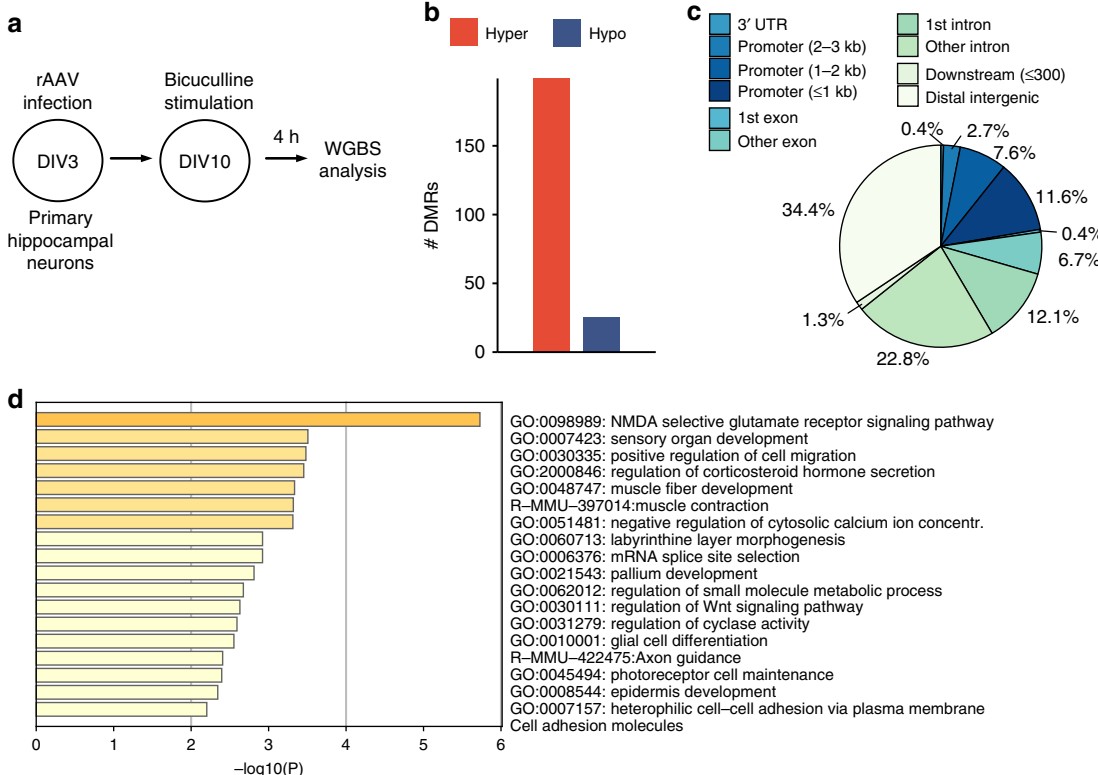

**Fig. 5 Dnmt3a2 alters the DNA methylation profile of memory-related genes. a** Experimental design used to identify DNA methylation changes in response to Dnmt3a2 overexpression in primary hippocampal cultures. **b** Number of hyper- (higher methylation in HA-Dnmt3a2-expressing cells) and hypo- (lower methylation in HA-Dnmt3a2-expressing cells) DMRs identified by whole-genome DNA methylation profiling. $N = 3$ independent cell preparations per condition (ESARE-HA-GFP versus ESARE-HA-Dnmt3a2). **c** Genomic distribution of DMRs with respect to the gene model. **d** Top GO categories found enriched using the Metascape analysis of genes in the vicinity of hyper-DMRs. Shown are only categories with adjusted $P > 0.01$ (hypergeometric test, Benjamini–Hochberg adjustment). DIV: day in vitro, WGBS: whole-genome bisulfite sequencing, DMR: differentially methylated regions, rAAV: recombinant adeno-associated virus, UTR: untranslated region.

regulated Dnmt3a2[12] specifically within learning-activated neurons and with kinetics similar to those achieved by the natively expressed gene[12].

Our results demonstrate that restricting the overexpression of Dnmt3a2 to the majority of the DG neuronal population (~75%) activated by CFC was sufficient to positively modulate the memory engram, and to specifically strengthen the memory encoding for the aversive context. Strikingly, Dnmt3a2 overexpression within the learning-activated neuronal ensemble improved its reactivation by recall without altering its overall size. Our findings thus implicate DNA methylation as a mechanism that regulates the strength of memory, likely by modulating the stability of the neuronal ensemble during consolidation and thereby increasing the fidelity of its reactivation during memory recall. Furthermore, our findings indicate that the strength of a memory is dictated not by the size of recall-induced neuronal population, but rather by the precision of engram reactivation. The stabilization of neuronal ensembles is possibly determined by a selective strengthening of the connectivity between members of the neuronal population activated by learning. Here we found that the expression of Dnmt3a2 triggered by neuronal activity in primary hippocampal cultures leads to DNA methylation changes preferentially in genes that regulate synaptic strength and structure and learning and memory. Namely, genes involved in the modification of the postsynaptic membrane (*Tiam1, Ctnnd2, Cabp1, Kalrn, Lrrtm1, Farp1*) and in neurotransmitter receptor activity (*Grin2b, Gabrg2*), several of which have demonstrated roles in memory formation (*Tiam1*:[23–26]; *Cabp1*:[27–29]; *Lrrtm1*:[30,31]). Thus, this observation suggests that the expression of Dnmt3a2 in behaviorally allocated

neurons modulates functional and structural plasticity in neuronal ensembles during memory consolidation, which likely strengthens engram stability and determines the likelihood of their reactivation. Furthermore, we found that the DNA methylation changes predominantly occurred in intergenic and intronic regions. These findings are in line with two previous studies that showed similar enrichment for activity-dependent DNA methylation changes[40,41]. Thus, indicating that the regulatory functions of DNA methylation in memory formation go beyond promoter methylation. DNA methylation at intergenic and intronic regions may regulate the neuronal transcriptomic profile through several mechanisms that include modulation of enhancer activity[40], alternative splicing[42] and the expression of non-coding RNAs[43]. Future studies will elucidate how the identified DNA methylation changes influence gene activity. We speculate that DNA methylation-dependent changes in the neuronal properties may impact not only recall-dependent reactivation of the neuronal ensemble, but also activity replay during rest phases or sleep after an experience. Studies in rodents and humans showed that replay activity correlates positively with successful memory retrieval[44–47]. It is plausible that DNA methylation-dependent plasticity within neuronal ensembles facilitates their co-activation during replay, leading to their strengthened inter-connectivity and consequently to improved memory retrieval. It has been demonstrated that the activity of the DG during memory consolidation is also necessary for the maturation and long-term maintenance of neuronal ensembles spanning multiple brain regions[21,48]. Thus, though not addressed in this study, it is possible that, increasing the stability of DG neuronal ensembles through Dnmt3a2 overexpression

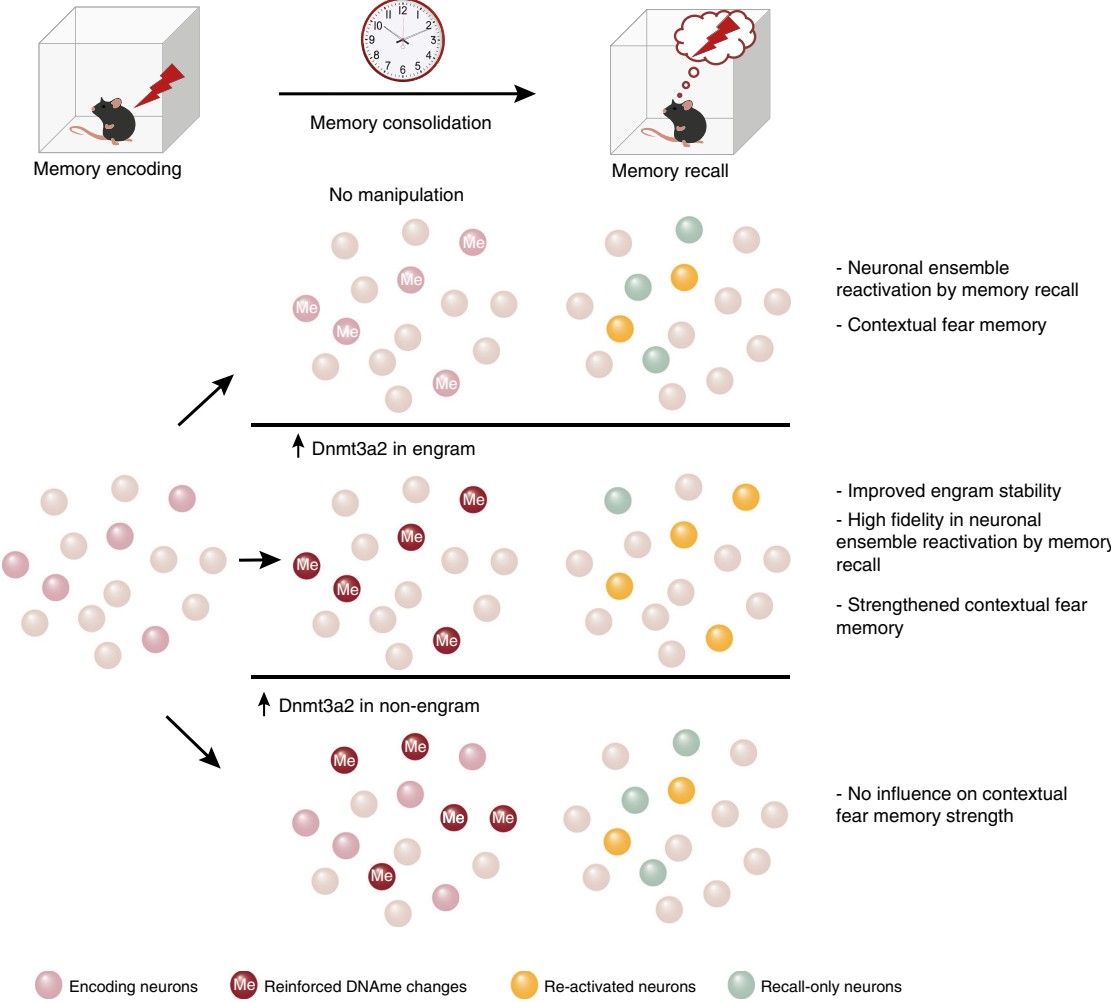

**Fig. 6 Graphical summary of the contribution of epigenetic mechanisms to hippocampal engram stability.** Reinforcing DNA methylation-related mechanisms through Dnmt3a2 overexpression within memory encoding DG neuronal ensembles (engram) during consolidation was sufficient to strengthen contextual fear memory and engram stability. This manipulation improved the reactivation of the encoding ensembles upon memory retrieval without altering the size of the neuronal population activated by recall. Whereas reinforcing DNA methylation-related mechanisms by overexpression of Dnmt3a2 in a similarly sized, but random subset of DG neurons (non-engram) during the consolidation window did not alter memory strength. Spheres represent DG neurons. Red color gradient in Me-labeled spheres represent the magnitude of DNA methylation-related changes. Me: DNA methylation-related changes.

might also facilitate systems-level consolidation and remote memory.

Our results attribute differential roles to DNA methylation writers prior to and after learning in the shaping of the memory engram. We found that increased Dnmt3a2 levels prior to learning do not bias the allocation of neuronal ensembles to the engram. In contrast, Dnmt3a2 overexpression in the neuronal population activated by learning impacted the engram. These findings suggest that Dnmt3a2 overexpression per se in a sparse population does not confer properties to the neurons that render them eligible to be recruited into an engram, but rather that Dnmt3a2 must be expressed in neurons that are responsive to a stimulus in order to regulate the memory engram. We found that the improved ensemble reactivation during memory retrieval was specific to the upper blade of the DG. Previous reports showed that, in rodents, the DG upper blade is more responsive to learning than the lower blade[49–52]. Different anatomical and physiological properties likely underlie this difference[49–52]. Thus, the anatomical specificity of our findings further corroborates the idea that Dnmt3a2 enhances engram stability when it works in conjunction with stimulus-induced mechanisms. A likely scenario

is that Dnmt3a2 participates in the regulation of stimulus-dependent transcriptional responses during the consolidation window[9]. Interestingly, previous studies that also manipulated the levels of a key memory molecule in a non-ensemble population prior to learning produced distinct findings[5,6]. In these studies, overexpression of cAMP response element-binding protein (CREB) in a sparse subset of lateral amygdala neurons before fear conditioning biased the recruitment of these neurons to a memory trace and promoted enhanced fear memory. Later on, it was shown by the same group that ensemble allocation is determined by the relative neuronal excitability immediately before training, a property that is modulated by CREB[7]. These findings together with our data, suggest distinct functions for CREB and Dnmt3a2 namely the ability to modulate neuronal properties in the absence of salient environmental stimuli.

Memory engram properties, such as stability, specificity, and dynamics have been associated with memory precision[21] and suggested to be compromised in aging and in pathological conditions such as Alzheimer's disease[17,53]. Indeed, deficits in these properties are correlated with impaired cognitive abilities[17,21,53]. In light of our findings, we suggest that targeted epigenetic

manipulations confined to specific neuronal ensembles may—by improving engram properties—provide a strategy to ameliorate cognitive dysfunction[54]. Moreover, in light of a recent study showing that efficient attenuation of fear memory requires reactivation of fear memory recall-induced neurons during extinction[55], we believe that, when applied during extinction learning, mechanisms that improve engram reactivation like the one found in our study, are likely to facilitate the attenuation of traumatic memories.

In summary, we performed for the first time in this study the targeted epigenetic manipulation of behaviorally allocated neuronal ensembles. We showed that the expression levels of a DNA methyltransferase within neuronal ensembles determine the precision and fidelity of ensemble reactivation and thereby also memory strength. Additionally, our results expand the current understanding of the functions of DNA methylation in cognitive abilities. By linking engram stability to the levels of an epigenetic factor, our findings set the stage for future investigations aimed at understanding the contributions of epigenetic changes to memory engrams. It is plausible that distinct epigenetic mechanisms within neuronal ensembles regulate transcriptional changes that underlie the structural and functional properties of this neuronal population. Although yet to be investigated, one can envisage that not only DNA methylation, but also demethylation, catalyzed by the Ten eleven translocation (TET) family members, and histone modifications play a critical role in the regulation of transcriptional changes within neuronal ensembles during memory consolidation. The interplay between these enzymes may dictate the permissiveness for gene transcription activation. Lastly, our findings underscore the importance of developing an improved understanding of the molecular mechanisms governing neuronal ensemble properties and their potential impairments under pathological conditions.

## Methods

**Animals**. Throughout the study, male C57BL/6N mice (Charles River, Sulzfeld, Germany) were used. The animals were housed on a 12 h light/dark cycle (06:00 to 18:00) and had access to water and food *ad libitum*. For the experiments using a doxycycline-dependent expression system, mice were kept on doxycycline-containing diet (40 mg/kg, BioServ, Flemington, NJ, USA) from the day of stereotaxic surgery. Two to three days prior to the experiment day, doxycycline food was replaced by regular food. Experiments were carried during the light-phase. All mice were 8–10 weeks old at the beginning of experiments. Mice that were sick and/or injured from cage-mate fighting were excluded from the study. All procedures took place according to the German guidelines for the care and use of laboratory animals (Regierungspraesidium, Karlsruhe, Germany) and with the European Community Council Directive 86/609/EEC.

**Recombinant adeno-associated virus**. rAAVs were produced by co-transfection of human embryonic kidney (HEK) cell line 293 (ATCC, Manassas, Virginia) with the target AAV plasmid and helper plasmids (pFΔ6, pRV1and pH21) using standard calcium phosphate precipitation. 60 h after transfection, HEK 293 cells were harvested and lysed. Finally, the viral particles were purified using heparin affinity columns (HiTrap Heparin HP; GE Healthcare, Uppsala, Sweden) and concentrated using Amicon Ultra-4 centrifugal filter devices (Millipore, Bedford, MA)[56]. For the overexpression of Dnmt3a2 or control gene expression (GFP or LacZ) within recently activated neurons, we generated the following rAAVs: E-SARE-HA-Dnmt3a2 (rAAV-E-SARE-HA-Dnmt3a2-WPRE-pA::hSynapsin-tDimer), E-SARE-HA-GFP (rAAV-E-SARE-GFP-HA-WPRE-pA::hSynapsin-tDimer) and E-SARE-HA-LacZ (rAAV-E-SARE-HA-LacZ-WPRE-pA::hSynapsin-tDimer). The E-SARE promoter was obtained from the plasmid pAAV-E-SARE-d2Venus::EPGK-FP635-WPRE[16] (a kind gift from Dr. H. Bito, Tokyo, Japan). To randomly overexpress Dnmt3a2 or express GFP in a sparse subset of neurons the following rAAVs were used: rAAV-CaMKIIα-Myc-CreER^T2-NES which contains a Cre recombinase fused to a mutated form of the ligand-binding domain of the estrogen receptor α (CreER^T2), a Myc-tag and a nuclear export sequence, rAAV-Synapsin-loxP-stop-loxP-GFP (loxP-GFP), and rAAV-Synapsin-loxP-stop-loxP-HA-Dnmt3a2 (loxP-HA-Dnmt3a2)[14]. For the tagging of neuronal ensembles in the retrieval-dependent reactivation rate analysis, the rAAV-RAM was used (pAAV-RAM-d2TTA::TRE-MCS-WPRE-pA was a gift from Yingxi Lin (Addgene plasmid # 63931; http://n2t.net/addgene:63931; RRID:Addgene_63931)[19]. A GFP

expression cassette containing a myc or HA tag was inserted into the multiple cloning site (MCS) of rAAV-RAM.

**Stereotaxic surgery**. rAAVs were injected into the dentate gyrus (DG) of the hippocampus at the following coordinates relative to Bregma: −2.0 mm anteroposterior, ±1.3 mm medio-lateral, −2.4 mm dorsoventral. For single injections, a total volume of 0.3 μl of virus was injected per hemisphere. For co-injections of E-SARE and RAM rAAVs, a total volume of 0.55 μl of virus (5:6 ratio of pRAM:E-SARE rAAVs) was injected per hemisphere. The injection speed was 80–100 nl/min through a 33-G beveled Nanofil needle (WPI, Sarasota, FL, USA). Before and after each infusion, the needle was left in place for 10 min to allow diffusion to the tissue and avoid spreading through the needle track that could result in infection of other hippocampal regions. Note that mice having unintended viral expression in the CA1 region of the hippocampus, or cortical regions were excluded from the analysis. After stereotaxic surgery until the end of the experiment and data analysis, the experimenter was blind to the identity of the virus injected into each mouse.

**Behavioral testing**. Two days after the stereotaxic surgery, mice were habituated to the experimenter and experiment room by gentle handling for 5 days, 2 min per mouse. Habituation was performed in the first week of rAAV delivery, when the rAAV expression is almost undetectable[14,20], to minimize the habituation-induced expression of Dnmt3a2. Two weeks after the last handling day, CFC or spatial object learning was performed[57]. Briefly, for CFC, mice were placed into the conditioning-chamber (23 × 23 × 35 cm, TSE, Bad Homburg, Germany) and 148 s later, received a 2 s foot shock (0.5 mA). 30 s after the shock terminated, mice were removed from the chamber. Spatial object learning consisted of four sessions (each session lasted 6 min and was separated by 3 min interval in the home-cage). In the first session, mice were placed in a black, square open field (50 cm × 50 cm × 50 cm) with a visual cue placed on the arena wall. In the next three sessions, two diagonally located distinct objects (a glass bottle and a metal tower) were present in the arena, which the mice explored freely. In case of the analysis of the overlap ratio between the behaviorally-allocated neuronal ensembles and E-SARE- or Cre- expressing neuronal ensembles after hippocampal learning, mice were sacrificed 2 h after the onset of spatial object learning or CFC, respectively. Otherwise, mice were returned to the housing room and were undisturbed until the memory test, 24 h later. During the memory test, the freezing behavior was scored manually either in the same conditioning-chamber or in an altered context. Altered context was designed as dissimilar as possible to the conditioning-chamber. The chamber was located in a different experimental room. Furthermore, floor (gray plastic instead of metal grid), scent (lemon detergent instead of ethanol), shape (triangle instead of square) and the light intensity was altered in the new chamber. The duration of the test was 5 min. In the experiments concerning the analysis of memory recall-induced reactivation of the initial ensembles, mice were sacrificed 2 h after the memory test onset. For the overlap between different ensemble marking tools (E-SARE and RAM systems), mice were perfused 24 h after the CFC onset, without undergoing a memory test.

**Immunohistochemistry**. Intracardiac perfusions of mice were performed with 4% paraformaldehyde (Sigma-Aldrich, Munich, Germany). Brains were collected carefully and further fixed in 4% paraformaldehyde (Sigma-Aldrich, Munich, Germany) overnight at 4 °C. Next day, brains were transferred to 30% sucrose solution in phosphate-buffered saline (PBS) for cryoprotection. Brain slices were cut at a thickness of 20 μm and permeabilized with 0.1% Triton X-100 in PBS. After blocking in 8% normal goat serum with 0.3% Triton X-100 in PBS for 50 min at room temperature, slices were incubated with primary antibody (diluted in PBS containing 2% normal goat serum and 0.3% Triton X-100) at 4 °C overnight. Next day, slices were incubated with secondary antibody (diluted in the same solution as primary antibody) for 2 h in the dark at room temperature. Lastly, slices were incubated in Hoechst 33258 (2 μg/ml, Serva, Heidelberg, Germany) for 5 min and mounted on glass slides. When required, slices were photobleached prior to immunohistochemistry by incubation with 5% hydrogen peroxide (Sigma-Aldrich, Munich, Germany) for 2 h under white light at room temperature. For immunostaining of 5mC, prior to the abovementioned immunohistochemistry protocol, brain slices were incubated in 1 M HCl incubation for 2 h at room temperature after permeabilization. Primary antibodies used in this study are as following: HA-tag (Covance, MMS-101R (1:1000); Roche, #1867423 (1:100)), Arc (Synaptic Systems, 156003 (1:1000)), Fos (Cell Signaling, 2250 (1:1000)), GFP (Aves Labs, GFP-1020 (1:1000)), 5mC (Active Motif, 39649 (1:250)).

**Hippocampal neuronal cultures**. Hippocampi of P0 C57Bl/6N mice were dissociated by papain digestion and plated onto tissue culture dishes coated with poly-D-lysine and laminin (Sigma-Aldrich, Munich, Germany). The primary cultures were maintained for 8 days in Neurobasal-A medium (Gibco™) supplemented with 1% rat serum (Biowest), 0.5mM L-glutamine (Sigma-Aldrich, Munich, Germany) and B27 (Gibco™), followed by incubation in salt-glucose-glycine solution (10 mM HEPES, pH 7.4, 114 mM NaCl, 26.1 mM NaHCO₃, 5.3 mM KCl, 1 mM MgCl₂, 2 mM CaCl₂, 30 mM glucose, 1 mM glycine, 0.5 mM $C_3H_3NaO_3$, and 0.001% phenol red) and phosphate-free Eagle's minimum essential medium (9:1 v/v), supplemented with insulin (7.5 μg/ml), transferrin (7.5 μg/ml), and sodium

selenite (7.5 ng/ml) (ITS Liquid Media Supplement, Sigma-Aldrich, Munich, Germany) and penicillin-streptomycin. Hippocampal cultures were treated with AraC (Sigma-Aldrich, Munich, Germany) at day in vitro 3 (DIV 3) and infected with rAAVs at DIV 4. Experiments were performed at DIV 10. To induce neuronal activity, the cultures were treated with Bicuculline at a concentration of 50 μM (Alexis Biochemicals, Farmingdale, NY, USA). To block neuronal activity, tetrodotoxin at a concentration of 0.5 μM was used (BioTrend, Zurich, Switzerland).

**Immunocytochemistry**. Primary hippocampal neurons plated on coverslips were rinsed with PBS and fixed with a prewarmed solution of 4% paraformaldehyde and 4% sucrose for 15 min at room temperature. After permeabilizing the cells in methanol for 6 min at −20 °C, blocking was performed with 10% normal goat serum in PBS for 1 h at room temperature. Next, cells were incubated with primary antibody (diluted in PBS containing 2% BSA, 0,1% Triton X-100) for overnight at 4 °C, which continued with secondary antibody incubation (diluted in PBS containing 2% BSA, 0,1% Triton X-100) for 1 h at room temperature. Finally, coverslips were treated with Hoechst 33258 (2 μg/ml, Serva, Heidelberg, Germany) for 5 min and mounted on glass slides. When required, slices were photobleached prior to immunocytochemistry by incubation with 5% hydrogen peroxide (Sigma-Aldrich, Munich, Germany) for 45 min under white light at room temperature. For immunostaining of 5mC, prior to the abovementioned immunocytochemistry protocol, coverslips were incubated in methanol for 6 min at −20 °C followed by 1 M HCl incubation for 2 h at RT. We used the following primary antibodies for immunocytochemistry in this study: HA-tag (Santa Cruz, sc805 (1:500)), 5-mC (Calbiochem, NA81 (1:500)).

**Western blotting**. Hippocampal neuronal cultures were harvested with 2× SDS-Sample buffer (160 mM Tris-HCl (pH 6.8), 4% SDS, 30% glycerol, 10 mM dithiothreitol, and 0.02% bromophenol blue). In the case of western blotting of tissue samples, DG of the hippocampus was dissected carefully from mouse brain and the tissue was homogenized in RIPA buffer (150 mM NaCl, 1% Triton X-100, 0.5% sodium deoxycholate, 0.1% SDS, 50 mM Tris, pH 8.0) supplemented with 1% proteinase inhibitor cocktail (Sigma-Aldrich, Munich, Germany). Protein concentration was measured by Bradford assay and 37.5 μg of protein was loaded on a 10–12% acrylamide gel after being denatured at 95 °C for 5 min. After blotting onto a nitrocellulose membrane (GE Healthcare, Buckinghamshire, UK), membranes were blocked in 5% milk in PBS with 0.01% Tween for 1 h at room temperature. Next, the membranes were incubated with primary antibodies (diluted in 5% milk in PBS with 0.01% Tween) for overnight at 4 °C, which was followed by horseradish peroxidase-conjugated secondary antibody incubation (diluted in 5% milk in PBS with 0.01% Tween) for 1 h at room temperature. Finally, blots were treated with western blotting detection reagent (GE Healthcare) and either exposed to Hyperfilm (GE Healthcare) using an X-ray processing machine or developed with ChemiDoc imaging system (Bio-Rad, Hercules, CA, USA). We used the following primary antibodies for western blotting in this study: HA-tag (Covance, MMS-101R (1:7500)), Arc (Synaptic Systems, 156003 (1:6000)), Alpha-tubulin (Sigma-Aldrich, T9026 (1:400,000)), hr-GFP (Stratagene, 240142 (1:20,000)), myc-tag (Santa Cruz, sc-40 (1:500)).

**Microscopy and image analysis**. For immunohistochemistry, images were acquired using Nikon A1R confocal microscopy (at Nikon Imaging Center, Bio-Quant, Heidelberg) at ×20 magnification (3 images per Z-stack; z-distance between two consecutive images: 2 μm) using NIS-Elements software. 15% overlap ratio was set to stitch individual tiles of a bigger image (typically 3–5 images per DG). Identical microscope settings (gain, offset, stitching overlap ratio) were used between different experimental groups and the experimenter was blind to the identity of images until the end of the image analysis. Z-stacks were imported in Fiji[58], background fluorescence of each stack was subtracted and a maximum projection file of each fluorescence channel in each stack was generated. On each maximum projection file, the granule cell layer of the dentate gyrus (DG) of the hippocampus was selected as the region of interest and the rest of the image was cleared. When analyzing a single-channel immunostaining (to detect a single protein in addition to Hoechst staining), a threshold was applied to the cleared maximum projection file and particles above the threshold, and larger than 50-pixel units were identified positive. Each positive signal was confirmed to be a cell based on Hoechst staining. In case of an ambiguous signal on the maximum projection file, individual frames of the stack were reviewed. For the overlap analysis of two signals, a co-immunostaining was performed in which each protein was detected in a different fluorescence channel. From the acquisition of Z-stacks at the microscopy to the generation of maximum projection file of the DG for each channel in Fiji, all steps were carried as explained above. Next, a threshold was applied to the signal in the first channel, and particles above the threshold, and larger than 50-pixel units were identified positive. These cells were labeled as "positive for the first channel" by cell-counter plugin of Fiji. Then, a threshold was applied to the signal in the second channel, and particles above the threshold, and larger than 50-pixel units were identified positive. These cells were labeled as "positive for the second channel" by cell-counter plugin of Fiji. Only the particles identified positive in both of the channels were labeled as "overlapping". To calculate the total number of cells (Hoechst$^+$ cells), the size of each DG was measured

and the cell count in each respective DG was obtained manually. Based on the correlation between the total area of the DG and the cell count, a formula was created (from at least 15 DG, $R^2 > 0.80$) and used to estimate the number of Hoechst$^+$ cells in each DG analyzed in that experiment. Note that the formula was recalculated every time when there was a change in the thickness of slices, or microscopy/staining conditions. At least 2–3 brain slices per animal and at least 3 animals per condition were used for the image analysis throughout the study. Quantification of each slice was performed individually, and average of the multiple slices was used as the quantified rate per animal. In order to avoid artifacts due to viral delivery conditions among different experimental batches, the rate of each animal was normalized to the mean rate of control group. We used the following formulas in this study to calculate reactivation rate[35] and modified the similarity index[22] for our context;

$$\text{Observed overlap}: \frac{(\text{GFP} + \text{Fos}+)}{\text{Hoechst}+} \times 100$$

$$\text{Overlap by chance}: \frac{\text{GFP}+}{\text{Hoechst}+} \times \frac{\text{Fos}+}{\text{Hoechst}+} \times 100$$

$$\text{Reactivation rate}: \frac{(\text{GFP} + \text{Fos}+)}{\text{GFP}+} \times 100$$

$$\text{Similarity index}: \frac{(\text{GFP} + \text{Fos}+)}{(\text{GFP}+) + (\text{Fos}+) - (\text{GFP} + \text{Fos}+)} \times 100$$

For the correlation analysis of HA-signal intensity and 5-mC signal intensity in primary hippocampal neurons, coverslips were imaged with a ×40 oil objective mounted on a fluorescence microscope (Leica Microsystems). Five different regions from each coverslip, and four coverslips from four independent primary hippocampal culture preparation batch per condition were imaged. Identical microscope settings were used between different experimental groups. Image quantification was performed using Fiji[58]. The background fluorescence was measured for each image and subtracted. Using analyze particles function of Fiji, integrated signal intensity (area × signal intensity) of each neuron was calculated in both of the channels. Integrated signal intensity of each channel was normalized to the mean of the signal intensities. Finally, normalized integrated signal intensities of each neuron for both channels were plotted for correlation analysis. For measuring 5-mC intensity in HA-Dnmt3a2-positive neurons versus HA-Dnmt3a2-negative neurons, a threshold was applied to the signal in the channel used to detect the HA-tag. Particles above the threshold, and larger than 50-pixel units were marked as region of interests (ROIs) in Fiji and identified as "HA-positive neurons". In addition to those neurons, 10–15 neurons that display HA-signal intensities below the applied threshold were randomly selected and marked as ROIs in Fiji, and identified as "HA-negative neurons". In order to obtain the 5-mC signal intensity of the depicted neuronal populations, marked ROIs in HA-channel were then projected to the channel used to detect 5-mC staining, and the mean 5-mC signal intensity of each ROI was measured. Average of the mean 5-mC signal intensity in all of the "HA-positive neurons" or "HA-negative neurons" in each brain slice was calculated. At least 2–3 brain slices per animal and at least 3 animals per condition were used for the final statistical comparison. Average of the multiple slices per animal was used as the rate per animal. In order to avoid artifacts due to viral delivery conditions among different experimental batches, mean 5-mC signal intensity of each group was normalized to the mean of the "HA-negative neurons" per animal.

**Genome-wide DNA methylation profiling using WGBS**. Genomic DNA was isolated from hippocampal cultures that were infected with ESARE-HA-GFP or ESARE-HA-Dnmt3a2 viruses and treated with bicuculline. The DNA isolation was performed using DNeasy Blood and Tissue kit (Qiagen, Hilden, Germany) according to the manufactures' instructions and approximately 600 ng per sample were used for WGBS analysis (three independent cell preparations were used per condition). WGBS libraries were prepared by the German cancer research center (DKFZ) Genomics and Proteomics Core Facility using the SWIFT Biosciences protocol, and submitted for sequencing on the Illumina X-Ten machine. Thereby generated sequence reads were submitted to the Omics IT and Data Management Core Facility and processed by the standardized bisulfite alignment and quality control workflow (https://github.com/DKFZ-ODCF/AlignmentAndQCWorkflows) using the mm10 mouse genome assembly as reference. Methylation calls were imported into R statistical environment and further analyzed using methrix package (https://github.com/CompEpigen/methrix). Differential methylation analysis was performed using DSS[59]. DMR annotation with respect to known mouse genes using the ChipSeeker[60] with the TxDb.Mmusculus.UCSC.mm10.knownGene database of murine transcripts available at Bioconductor. Functional annotation of DMR-associated genes was performed using Metascape with default parameters[61] using all DMR-gene associations defined by ChipSeeker, including closest genes from the Distal Intergenic category.

**Data and statistical analysis**. The behavioral data were collected from at least three independent sets of experiments. Each set of experiments contained mice injected with control or experimental rAAVs. Experimenter was blind to the identity of viruses injected to each mouse until the end of the experiment. Each

data set was subjected to a normality test prior to further comparisons (Shapiro-Wilk normality test; alpha = 0.05). For normally distributed data sets, when comparing two independent samples (i.e., data points from different subjects) two-tailed unpaired Student's t-test, whereas when comparing two dependent samples (i.e., data points from the same subject) two-tailed paired t-test was used. For non-normally distributed data sets, when comparing two independent samples (i.e., data points from different subjects) two-tailed Mann–Whitney test, whereas when comparing two dependent samples (i.e., data points from the same subject) two-tailed Wilcoxon matched-pairs signed-rank test was used. For correlation analysis, Pearson correlation test was applied. All of the statistics throughout the paper were performed using GraphPad Prism for Mac OS X, version 7.

**Reporting summary**. Further information on research design is available in the Nature Research Reporting Summary linked to this article.

## Data availability

The source data underlying Figs. 1b–d, 2b, f, g, 3c–e, 4c–e and Supplementary Figs. 1c, e, 2b, d–g, 3a, b, 4b–e, 5b–d, 6c, d, 7a–c, e, f, 8a–c are provided as a Source Data file. The WGBS data that support the findings of this study are available in European Nucleotide Archive (ENA) with the accession code PRJEB34831.

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

## Acknowledgements

We thank I. Bünzli-Ehret for the preparation of primary hippocampal cultures and Dr. Anna Hagenston Hertle for her critical comments to the paper. We thank the High Throughput Sequencing unit of the Genomics & Proteomics Core Facility, German Cancer Research Center (DKFZ) for the WGBS services and the support by the Omics IT and Data Management Core Facility (ODCF) of the DKFZ. This work was supported by the Deutsche Forschungsgemeinschaft (DFG) (Sonderforschungsbereich (SFB) 1134 (project C01) (to A.M.M.O.), an Emmy Noether grant (OL 437/1-1) (to A.M.M.O.) and the Helmholtz Foundation (to C.P.). A.M.M.O. is a member of the Excellence Cluster CellNetworks at Heidelberg University.

## Author contributions

A.M.M.O. conceived the project. K.G.K. and A.M.M.O. designed the experiments. K.G.K., J.K. and D.V.C.B. performed experiments and analyzed the data. B.Z. produced the viruses and provided technical assistance throughout the project. C.T. performed experiments and analyses during the revision stage of the study. D.W., P.L. and C.P. performed the analysis of the WGBS and interpreted the data. K.G.K. and A.M.M.O. wrote the paper. All authors revised the paper.

## Competing interests

The authors declare no competing interests.

## Additional information

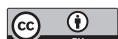

