## [Peer Review File · Nature Communications]

Reviewers' comments:

Reviewer #1 (Remarks to the Author):

The study by Karaca and colleagues examines the role of the DNA methyltransferase Dnmt3a2 in neuronal ensembles activated by contextual fear conditioning. The approach involved a variety of viral and genetic methods to express Dnmt3a2 from an activity-dependent promoter (ESARE). The main findings suggest that Dnmt3a2 overexpression can enhance memory, but sparse expression fails to do so, and that re-exposure to the conditioned context to engage retrieval mechanisms preferentially engages a Dnmt3a2 overexpressing ensemble. Overall, the results are interesting as they begin to implicate a powerful epigenetic mechanism in 'neuronal ensemble' activity during encoding and retrieval. However, even though the study employs very sophisticated approaches, there are some interpretation issues.

1- What was the level of expression of HA-Dnmt3a2 compared to endogenous levels of Dnmt3a2?

2- It's not clear whether the level of overexpression of Dnmt3a2 significantly affects DNA methylation. The evidence that the approaches used here affect DNA methylation in a meaningful manner (e.g. at IEGs to correlate with the results of the study) is missing. Without that data, there are several other interpretations that could be derived from the results. One of the main concerns is whether the activity of Dnmt3a2 is really necessary for the observed effects on enhanced memory in Figure 1?

3- The ESARE system appears to be leaky. There is HA-Dnmt3a2 expression at the zero time point in Supplementary Figure 1b. Does the same happen in vivo (not cell culture)? If so, that may alter the interpretation of the results. Results from Han et al (2007) suggest that simple overexpression of CREB in amygdala neurons is sufficient to recruit those neurons to be engaged during learning. It is a subtle but important point. The authors write that overexpression strengthens fear memory, but perhaps the leaky expression preferentially drives those neurons to be engaged. Thus, if there are more neurons overexpressing Dnmt3a2 because the virus hit more neurons than would normally be engaged during contextual fear conditioning, then there is an enhancement in memory observed.

4- The results shown in Figure 2 are potentially interesting, yet could also be interpreted in a different way. The authors conclude that together with data from Figure 1, that Dnmt3a2 overexpression must be confined to the behaviorally allocated neuronal ensemble in order to dictate memory strength. That may be true if Dnmt3a2 overexpression in a non-ensemble set of neurons had no effect, and that the non-ensemble set of neurons is similar in number of neurons, type of neuron, etc. In Figure 2 there are very few neurons with overexpression, leading to a sparse expression pattern that cannot be directly compared to figure 1 data.

5- In addition, do the two approaches in Figure 1 and Figure 2 lead to expression in the same types of neurons in the DG? This also affects interpretation of the results and trying to compare them.

6- The titles of the sections are a little misleading. For example, one says 'increased DNA methylation prior to learning does not alter the allocation of neurons into the engram'. Yet changes in DNA methylation were not examined in that experiment.

Reviewer #2 (Remarks to the Author):

General:

The authors address the question of the molecular mechanism of memory consolidation and the role of DNA methylation in this process. The question is of high interest to the neuroscience community and more generally, characterization of the contribution of DNA methylation to function and behaviors in an adult is of great interest. The authors use activity dependent promoters to overexpress Dnmt3a2 in recently activated neurons (engram cells) in the DG during a learning task, which enables testing the contribution of Dnmts to memory consolidation and recall. Specifically, they use various manipulations of specific promoters and viruses to show that this effect is dependent on specific cells, timing, and also has a longer effect on recall. The discovery of epigenetic regulation of memory formation and recall is meaningful; however, I have several concerns regarding the robustness of the results. In addition, molecular evidence (e.g. methylation, expression) indicating on the downstream mechanism involved will highly improve the results and advance toward a mechanism.

Specific comments:

1. Expression levels of Dnmt3a. In supplementary Figure 1b we see low levels of HA-Dnmt3a detected in time 0 (and not detected for Arc or GFP). This issue should be addressed since it can result from several causes each affecting the results. Specifically, this time 0 expression could be due to activation during habituation time and slow degradation of the protein compared to the GFP or rather an indication of leaky expression? Related to this issue, a longer quantification of the HA-Dnmt3a should be added to show how long does the protein stay after activation.
2. Sparse activation of Dnmt3a. It seems that the sparse activation reaches overall lower levels of the protein or fewer neurons that express the gene (based on Figure 2, S3, and in comparison, to Figure S1). This presents an alternative explanation to the lack of effect in recall compared to specific activation of Dnmts in engram cells (Figure 2), and also to the effect of expression prior to learning (Figure 3). Please provide additional evidence that the expression level or sparseness are not the limiting factors.
3. Statistical tests. Multiple tests are used in the significance test of overlaps and behavioral tests (for T-test you use both unpaired and paired, for non-normally distributed you use Mann-Whitney, and in addition you use Wilcoxon test). Please be more consistent with the statistical testing or explain better the rationale. Specifically refer to the following panels, where the consistent change in the results do not match the non-significance level measured: Figure S7c, Figure 3e, Figure S6e, Figure 4c.
4. The results suggest that over expression of Dnmt3a in a subset of engram cells affects memory and recall. However, the mechanisms through which the effect is mediated is not clear.
 - a. In the discussion you claim that it is: "likely by modulating the stability of the neuronal ensemble", but this claim is speculative and thus should be further explained or supported.
 - b. Evidence for DNA methylation or gene expression changes following the over expression of Dnmt3a will strengthen the manuscript significantly (global measurements are preferred but specific measurements of genes/domains previously suggested to change with learning/activity will also be helpful, see for example the work of Guo et al. 2011).
5. The % of activated neurons vary significantly between experiments and experimental systems (e.g. %Fos+ 0.5% (Figure S6), %HAGFP 2% (Figure S4)), which is concerning regarding the robustness of the system and the labeling of the true engram cells. How do you reconcile this? Is this technical or biological variation?
6. Points need to be addressed in the text/discussion -
 - a. Please address the fact that you over express the gene only in a sub-set of the engram cells.
 - b. Why did you choose to manipulate dnmt3a and not one of the other Dnmts or TET genes and how do you expect these other proteins to be involved.
 - c. Please explain your results in connection to previous contradictory results such as Morris et al. 2014 and Feng et al. 2010 each suggesting different roles for Dnmt3a/Dnmt1 in learning.

Additional minor comments:

1. Supplementary Figure 1f: please also show the actual numbers of double and single positive counts and note how many sections were used.
2. Please make sure all abbreviations in figures are explained in figure legends (e.g. CFC).
3. Please add graphs showing quantification of behavior (such as % time freezing) displaying the

actual data per mice (in addition to the average shown as bar plots).

4. Figure S4d: columns are not labeled in a coherent way, specifically what's the difference between the first three and second three (on/off Dox?)?

5. Figure 5. the summary figure is very clear and helpful. But some fonts are too small, and the meaning of NE is unclear.

6. Supplementary data tables, with the raw data from the behavior and image analysis would help improve the transparency and avoid confusion such as the one suggested above with the statistical testing.

7. Please provide more details regarding the image analysis or the code used to quantify the proteins and their overlap.

Reviewer #3 (Remarks to the Author):

The authors developed elegant viral manipulations to investigate the role of increased expression of a DNA methyltransferase (Dnmt3a2) in engram cells after contextual fear conditioning leading to enhanced memory formation. Overall, they come to the conclusion that increased Dnmt3a2 expression in engram cells in dentate gyrus would not affect engram size but rather enhance engram stability assuring improved retrieval. In my opinion this work is of very high impact in the field due to the novel approach to manipulate engram cells at molecular level and because of the suggestion that memory enhancement might be due to increased engram stability. However, I think that the manuscript would benefit from some more control experiments as outlined below:

1) Fig. S1b shows that the Dnmt3a2 overexpression system is leaky in primary neurons. Such leakiness is a great concern, when claiming to study the effect of Dnmt3a2 in engram cells. I think that the authors should characterise the leakiness in vivo. Importantly, they should quantify the leaky expression in dentate gyrus and compare with expression levels of the intended leaky expression system in Figure 2.

2) Fig. S1c indicates that the viral transfection is not only in the dentate gyrus, but also other regions in the hippocampus, including area CA1. Therefore, the authors should analyse effects in area CA1 to assure that primarily dentate gyrus and not other hippocampal regions were manipulated.

3) It would be good to illustrate the kinetics of the inducibility of increased Dnmt3a2 expression after CFC to understand which consolidation window may be affected.

Thank you for your e-mail of April 30, 2019 regarding our manuscript entitled “*Neuronal ensemble-specific DNA methylation strengthens engram stability*” (NComms-19-06844). We are now submitting the revised version of the manuscript. New experiments have been done to address the comments of the referees. In particular, we added several experiments to characterize in more detail the expression systems, performed whole genome bisulfite analysis to identify DNA methylation changes driven by Dnmt3a2 ectopic expression and expanded the description of methodological aspects (image analysis and statistical analysis). The revised manuscript contains both new and substantially revised figures as well as textual revisions (marked in red in the manuscript).

We are also submitting a source data file that includes the source data underlying Figs 1b-d, 2b,f,g, 3c-e, 4c-e and Supplementary Figs 1c,e, 2b,d-g, 3a,b, 4b-e, 5b-d, 6c,d, 7a-c,e,f, 8a-c.

Our detailed responses to the Reviewers’ comments are as follows:

We would like to thank the editor and reviewers for their positive feedback and insightful suggestions on how to strengthen our manuscript.

Reviewers' comments:

Reviewer #1 (Remarks to the Author):

The study by Karaca and colleagues examines the role of the DNA methyltransferase Dnmt3a2 in neuronal ensembles activated by contextual fear conditioning. The approach involved a variety of viral and genetic methods to express Dnmt3a2 from an activity-dependent promoter (ESARE). The main findings suggest that Dnmt3a2 overexpression can enhance memory, but sparse expression fails to do so, and that re-exposure to the conditioned context to engage retrieval mechanisms preferentially engages a Dnmt3a2 overexpressing ensemble. Overall, the results are interesting as they begin to implicate a powerful epigenetic mechanism in ‘neuronal ensemble’ activity during encoding and retrieval. However, even though the study employs very sophisticated approaches, there are some interpretation issues.

1- What was the level of expression of HA-Dnmt3a2 compared to endogenous levels of Dnmt3a2?

In order to address this point we performed qRT-PCR analysis of hippocampal cultures treated with bicuculline to trigger the expression of both endogenous and exogenous *Dnmt3a2*. To identify the endogenous *Dnmt3a2* we used a Taqman probe that binds to the 5’UTR region of the *Dnmt3a2* transcript (the use of a probe against the coding sequence would not allow discriminating between the isoforms *Dnmt3a1* and *Dnmt3a2*). To quantify the exogenous *Dnmt3a2* we used a Taqman probe that binds to the WPRE sequence that is unique to *HA-Dnmt3a2* and is not present in any mouse transcript. We next confirmed that the probes have similar amplification efficiencies ($E_{(Dnmt3a2\ taqman\ probe)}=1.941 \pm 0.045$; $E_{(WPRE\ taqman\ probe)}=1.947 \pm 0.012$) and therefore can be used to directly compare the abundance of each transcript. We found that the expression of *HA-Dnmt3a2* is 14×10^4 -fold higher than endogenous *Dnmt3a2* in primary hippocampal cultures treated with bicuculline. It should be noted that the levels of expression of activity-regulated genes triggered by

a non-physiological form of synaptic activity, like the one obtained by bicuculline treatment, is certainly higher than a physiological stimulus, such as a brief learning episode in contextual fear conditioning. The continuous burst of activity in response to bicuculline likely causes accumulated, enhanced expression levels of activity-dependent genes in cultured neurons. Therefore, the expression levels of *HA-Dnmt3a2* compared to endogenous *Dnmt3a2* are likely higher in these conditions than what would occur *in vivo* in response to a physiological stimulus.

However, despite the different nature of stimulation, we chose to perform qRT-PCR analysis in primary hippocampal cultures rather than *in vivo* (dentate gyrus) for the following reasons: in culture conditions, the vast majority of the neuronal population (80 to 90%) expresses the exogenous *Dnmt3a2*. This is in contrast to *in vivo* in which the expression of *HA-Dnmt3a2* is triggered only in ~5% of the dentate gyrus granule cells. Therefore, the analysis of dentate gyrus tissue would dramatically underestimate the proportion of exogenous versus endogenous *Dnmt3a2* expression per cell. Moreover, single cell analysis by immunohistochemistry would not be possible: *Dnmt3a2* does not contain any unique stretches of amino acid sequences that allow distinguishing it from *Dnmt3a1* by the use of an antibody. Despite these limitations we could show that dentate gyrus granule cells that express *HA-Dnmt3a2* exhibit significantly higher levels of global DNA methylation compared to neighboring *HA-Dnmt3a2*-negative neurons (see new **Supplementary Figure 4b,c**). This shows that the levels of overexpression of *Dnmt3a2* significantly affect DNA methylation.

2- It's not clear whether the level of overexpression of Dnmt3a2 significantly affects DNA methylation.

Please see comment above.

The evidence that the approaches used here affect DNA methylation in a meaningful manner (e.g. at IEGs to correlate with the results of the study) is missing. Without that data, there are several other interpretations that could be derived from the results. One of the main concerns is whether the activity of Dnmt3a2 is really necessary for the observed effects on enhanced memory in Figure 1?

We agree with the reviewer that the identification of DNA methylation changes driven by the approaches used in our study (activity-dependent *Dnmt3a2* overexpression) adds significant value to the interpretation of our findings. Therefore to address this point we performed an unbiased analysis of DNA methylation changes using whole genome bisulfite sequencing in primary hippocampal cultures that expressed E-SARE driven GFP or *HA-Dnmt3a2* upon neuronal activity (new **Figure 5**). This experiment allowed us to identify the differentially methylated genomic regions and genes (DMRs and DMGs, respectively) in the condition of *Dnmt3a2* overexpression versus the control condition (new **Supplementary Table 1**). Interestingly, we found a significant overrepresentation of DMGs involved in synaptic plasticity and memory mechanisms (new **Supplementary Table 2**). Hence, this new data suggests that epigenetic regulation of memory-related genes underlies the observed effects. Furthermore, we found that the DNA methylation changes predominantly occurred in intergenic and intronic regions. These findings are in line with two previous studies that showed similar enrichment for activity-dependent DNA methylation changes^{1,2}. Thus, confirming that the regulatory functions of DNA methylation in memory go beyond a direct regulation of promoter activity. Future studies will elucidate how the identified DNA methylation changes influence gene activity. We thank the reviewer for the suggestion, as we believe that this experiment has significantly strengthened our study.

Although it would have been interesting to identify DNA methylation changes *in vivo*, that experiment would involve the selective analysis of DNA methylation in the very sparse dentate gyrus neuronal ensemble population. This would require extensive methodological establishment and characterization, in addition to the use of several additional animals. Therefore we think this is beyond the scope of the current revision and will be more suitable for future studies.

3- The ESARE system appears to be leaky. There is HA-Dnmt3a2 expression at the zero time point in Supplementary Figure 1b. Does the same happen in vivo (not cell culture)? If so, that may alter the interpretation of the results. Results from Han et al (2007) suggest that simple overexpression of CREB in amygdala neurons is sufficient to recruit those neurons to be engaged during learning. It is a subtle but important point. The authors write that overexpression strengthens fear memory, but perhaps the leaky expression preferentially drives those neurons to be engaged. Thus, if there are more neurons overexpressing Dnmt3a2 because the virus hit more neurons than would normally be engaged during contextual fear conditioning, then there is an enhancement in memory observed.

In the previous version of the manuscript, we characterized the E-SARE system *in vivo* by assessing GFP expression driven by the E-SARE promoter in basal conditions, after kainic acid administration or exposure to a novel environment (**Supplementary Figure 2a-e**). We agree with the reviewer that in addition to the analysis of GFP, HA-Dnmt3a2 expression *in vivo* should be performed. We have now analyzed the expression of HA-Dnmt3a2 in home cage conditions and after fear conditioning (**Supplementary Figure 2e,f**). We found that HA-Dnmt3a2 is expressed in basal conditions in 3,7% of the total dentate gyrus granule cell population. Our data shows that the basal expression of endogenous activity-regulated genes such as *Arc* occurs in 1% of the dentate gyrus (**Supplementary Figure 2b**). This indeed indicates that E-SARE-driven expression presents some baseline leakiness that is comparable to other studies that used neuronal ensemble labeling tools (~3%^{3,4}). Similar to the approach taken by Han et al (2007) we overexpressed Dnmt3a2 in a random and small population of neurons, in our case in ~5% of the total dentate gyrus granule cell layer (note that this comprises an even bigger population than the basal E-SARE-driven expression (3,7%)) (**Figure 2; Supplementary Figure 2e; Supplementary Figure 4a; page 7, lines 170,171**). We found that in contrast to CREB overexpression⁵, the non-ensemble overexpression of Dnmt3a2 did not lead to memory enhancement or to neuronal ensemble allocation (**Figure 2**). Therefore, this indicates that the slight baseline leakiness of the E-SARE system should not drive the HA-Dnmt3a2-positive neurons to be engaged in the memory engram or promote memory enhancement.

4- The results shown in Figure 2 are potentially interesting, yet could also be interpreted in a different way. The authors conclude that together with data from Figure 1, that Dnmt3a2 overexpression must be confined to the behaviorally allocated neuronal ensemble in order to dictate memory strength. That may be true if Dnmt3a2 overexpression in a non-ensemble set of neurons had no effect, and that the non-ensemble set of neurons is similar in number of neurons, type of neuron, etc. In Figure 2 there are very few neurons with overexpression, leading to a sparse expression pattern that cannot be directly compared to figure 1 data.

In order to address this point we have now quantified the E-SARE-driven expression of HA-Dnmt3a2 in home cage and triggered by fear conditioning (new **Supplementary Figure 2e,f**). The fear conditioning-driven expression of HA-Dnmt3a2 occurs in ~5% of the dentate gyrus cell population. Importantly, the size of the random non-ensemble HA-Dnmt3a2 expressing population is also ~5% (**Supplementary Figure 4a; page 7, lines 170-173**). Therefore the size similarity between the neuronal populations that overexpress Dnmt3a2 in a random manner or

in behaviorally allocated neurons should allow a comparison of the behavioral effects obtained by each manipulation. We thank the reviewer for this comment. It has led us to include in the manuscript a more appropriate comparison: the size of the HA-Dnmt3a2-expressing neuronal population in behaviorally allocated neurons (**Figure 1**) versus the random non-ensemble conditions (**Figure 2**).

5- In addition, do the two approaches in Figure 1 and Figure 2 lead to expression in the same types of neurons in the DG? This also affects interpretation of the results and trying to compare them.

The expression of HA-Dnmt3a2 in Figure 2 is controlled by the CaMKIIa promoter, therefore it is confined to excitatory neurons. The expression of HA-Dnmt3a2 in Figure 1 is regulated by the E-SARE promoter. Previous studies from the H. Bito laboratory confirmed that E-SARE-dependent expression is restricted to neurons⁶. We now attempted to characterize the proportion of excitatory versus inhibitory neurons that express E-SARE-driven HA-Dnmt3a2. However, in agreement with the literature⁷, the identification of inhibitory neurons in the dentate gyrus has proven very challenging. We don't feel confident that Gad67-positive cells can be identified with accuracy:

However, several aspects point to the conclusion that the vast majority of behaviorally allocated E-SARE driven HA-Dnmt3a2-expressing neurons (>90%) are excitatory:

- The E-SARE promoter that consists of the *Arc* gene regulatory regions controls HA-Dnmt3a2 expression. It has been shown that *Arc* expression in the hippocampus, driven by novel environment exposure, occurs in CaMKIIa-positive neurons but not in inhibitory neurons⁸.
- The Allen brain atlas and Gad67 reporter mouse lines⁹ show that in the dentate gyrus Gad67-positive neurons are mostly located in the deep surface of the granule cell layer (GCL) and hilus. We found that the vast majority of HA-Dnmt3a2-positive neurons are located in the granule cell layer (% HA⁺ neurons in the dentate gyrus GCL: 88.3 ± 2.3 , n=3; in the dentate gyrus hilus: 11.7 ± 2.3 , n=3) indicating that they are predominantly excitatory neurons.

- Moreover, among the neurons that we could identify with confidence as Gad67-positive, only a small proportion was positive for HA staining ($11.2\% \pm 2.2$, $n=3$).

For all these reasons we believe that the comparison between the results obtained in Figures 1 and 2 is valid.

6- The titles of the sections are a little misleading. For example, one says 'increased DNA methylation prior to learning does not alter the allocation of neurons into the engram'. Yet changes in DNA methylation were not examined in that experiment.

We agree with the reviewer and have now changed the titles to clearly express that the manipulation consisted in the overexpression of Dnmt3a2:

Previous: "Manipulation of DNA methylation in dentate gyrus neuronal ensembles."

New: "Manipulation of the levels of a *de novo* DNA methyltransferase in dentate gyrus neuronal ensembles."

Previous: "DNA methylation within neuronal ensembles strengthens fear memory in a context- and engram-specific manner."

New: "Dnmt3a2 overexpression within neuronal ensembles strengthens fear memory in a context- and engram-specific manner."

Previous: "Increased DNA methylation prior to learning does not alter the allocation of neurons into the engram."

New: "Increased Dnmt3a2 overexpression prior to learning does not alter the allocation of neurons into the engram."

Reviewer #2 (Remarks to the Author):

General:

The authors address the question of the molecular mechanism of memory consolidation and the role of DNA methylation in this process. The question is of high interest to the neuroscience community and more generally, characterization of the contribution of DNA methylation to function and behaviors in a adult is of great interest. The authors use activity dependent promoters to overexpress Dnmt3a2 in recently activated neurons (engram cells) in the DG during a learning task, which enables testing the contribution of Dnmts to memory consolidation and recall. Specifically, they use various manipulations of specific promoters and viruses to show that this effect is dependent on specific cells, timing, and also has a longer effect on recall. The discovery of epigenetic regulation of memory formation and recall is meaningful; however, I have several concerns regarding the robustness of the results. In addition, molecular evidence (e.g. methylation, expression) indicating on the downstream mechanism involved will highly improve the results and advance toward a mechanism.

Specific comments:

1. Expression levels of Dnmt3a. In supplementary Figure 1b we see low levels of HA-Dnmt3a detected in time 0 (and not detected for Arc or GFP). This issue should be addressed since it can result from several causes each effecting the results. Specifically, this time 0 expression could be due to activation during habituation time and slow degradation of the protein compared to the GFP or rather an indication of leaky expression?

We would like to mention that there was a mistake in the cropping of the Western blot of previous Figure S1b. As shown in the uncropped picture (source data file: Sup. Fig 1c), two bands are present in the lanes that correspond to the E-SARE-HA-GFP samples. Based on the expected protein size (28 KDa), the upper band should be considered. We apologize for this mistake and have now corrected it in the revised version of the manuscript (**Supplementary Figure 1c**). This picture shows that similarly to HA-Dnmt3a2, low levels of HA-GFP are detected at time 0. Two factors likely contribute to the expression of E-SARE-driven proteins at time 0; one, the basal neuronal activity in hippocampal cultures and second, some leakiness of the system.

In order to address the levels of HA-Dnmt3a2 *in vivo*, we have now also analyzed the expression of E-SARE-driven HA-Dnmt3a2 both in home cage conditions and after fear conditioning (**Supplementary Figure 2f,g**). We found that in home cage conditions, 3,7% of the dentate gyrus granule cells express HA-Dnmt3a2. The expression of the endogenous activity-regulated protein Arc in the same conditions is confined to 1% of the cells, thus indicating some leakiness of the system. This level of exogenous expression at basal conditions is similar to other studies that used neuronal ensemble labeling tools (3%^{3,4}). However, our study shows that this level of HA-Dnmt3a2 expression in basal conditions does not underlie the observed memory enhancement. More specifically, we showed that overexpression of HA-Dnmt3a2 in a random non-ensemble population of even slightly bigger size (5%) does not select the neurons that are recruited to the memory ensemble or affect memory performance (Figure 2, see also comment to point 2 for further details).

Lastly, the schedule of the behavioral experiments was designed to minimize unspecific HA-Dnmt3a2 expression at the time of the contextual fear conditioning. The mice were habituated to the experimental room and the experimenter for 5 consecutive days starting two days after stereotaxic surgery. We previously confirmed the absence of viral expression in the brain within the first week after the stereotaxic delivery of rAAVs¹⁰. Therefore, our strategy of habituating the mice during the period where rAAV expression was negligible, avoided habituation-related increase in the basal expression levels of E-SARE-driven proteins. Moreover, after the 5-day habituation period, the mice remained undisturbed in the housing room for two weeks until performing fear conditioning. In light of our new experiments that assess the duration of the protein expression after activation (**Supplementary Figure 1d,e** - we observed that the protein levels are significantly lower and close to baseline levels by 48 to 72h, see also comment below), a 2-week waiting period should guarantee that any expression triggered by the habituation was back to baseline.

Related to this issue, a longer quantification of the HA-Dnmt3a should be added to show how long does the protein stay after activation.

As suggested by the reviewer we have now performed a longer quantification of the HA-Dnmt3a2 expression (**Supplementary Figure 1d,e**). In order to determine how long does the protein stay after activation, we stopped action potential bursting through the incubation with tetrodotoxin (TTX) four hours after bicuculline treatment. This way we prevented the continuous synthesis of new transcripts in response to prolonged increased neuronal activity. We observed that by 48 to 72h the levels of HA-Dnmt3a2 or HA-GFP decayed significantly and are close to basal amounts. This indicates that the two exogenous proteins have comparable degradation kinetics.

2. Sparse activation of Dnmt3a. It seems that the sparse activation reaches overall lower levels of the protein or fewer neurons that express the gene (based on Figure 2, S3, and in

comparison, to Figure S1). This presents an alternative explanation to the lack of effect in recall compared to specific activation of Dnmts in engram cells (Figure 2), and also to the effect of expression prior to learning (Figure 3). Please provide additional evidence that the expression level or sparseness are not the limiting factors.

We thank the reviewer for bringing up this point for further clarification. To properly compare the ensemble-specific (E-SARE strategy) and random non-ensemble (Cre-loxP strategy) HA-Dnmt3a2 expression, we have now performed the analysis of brain slices from mice injected with rAAV-E-SARE-HA-Dnmt3a2 that were maintained in their home cage or trained in the fear conditioning task (note that before the characterization of E-SARE-dependent expression *in vivo* was done in slices from mice injected with E-SARE-HA-GFP in home cage conditions or exposed to a novel environment – previous Figure S1). We analyzed the number of neurons that express HA-Dnmt3a2 and respective expression levels. Moreover, we assessed the global DNA methylation levels in HA-Dnmt3a2-positive versus neighboring HA-Dnmt3a2-negative neurons in both ensemble-specific and random non-ensemble Dnmt3a2 overexpression strategies. We found that ensemble-specific expression of HA-Dnmt3a2 triggered by fear conditioning occurs in ~5% of the dentate gyrus granule cells (**Supplementary Figure 2e**). Importantly, random non-ensemble HA-Dnmt3a2 is confined to the same proportion of neurons (5%) (**Supplementary Figure 4a, page 7 lines 170-173**). This confirms that the two models exhibit equally sparse Dnmt3a2 overexpression. Concerning protein levels, we observed that the levels of ensemble-specific Dnmt3a2 expression are at least 2.5-fold higher than in non-ensemble. Despite this difference in expression levels, we verified that, similarly to the E-SARE system, the levels of HA-Dnmt3a2 in non-ensemble neurons, are enough to elicit significant changes in the global DNA methylation levels of the neurons (**Supplementary Figure 4b,c**). This result together with our findings in the reactivation studies (**Figures 3 and 4**) support that the different levels of HA-Dnmt3a2 expression between the two systems are unlikely to account for the different results. Concerning the experiment in Figure 3, we observed that HA-Dnmt3a2 is expressed at equally high levels in the upper and lower blades of the dentate gyrus (mean HA-Dnmt3a2 intensity (a.u.) in dentate gyrus upper blade=1473 +/- 57.24, in dentate gyrus lower blade= 1513 +/- 52.26; p=0.6726 by two-tailed paired t-test; n=4 mice), however an improved reactivation rate is only found in the upper blade (**Figure 3c-e**), indicating that high levels of HA-Dnmt3a2 expression do not *per se* determine improved reactivation. Moreover, if high levels of HA-Dnmt3a2 would be enough to bias ensemble allocation, one would expect that in the experiment in Figure 4 the expression of HA-Dnmt3a2 in neurons activated by context A, would bias the activation of the same neuronal population when mice were exposed to context B, which was not the case (**Figure 4**). For all these reasons we believe that the levels of HA-Dnmt3a2 are not the determining factor.

3. Statistical tests. Multiple tests are used in the significance test of overlaps and behavioral tests (for T-test you use both unpaired and paired, for non-normally distributed you use Mann-Whitney, and in addition you use Wilcoxon test). Please be more consistent with the statistical testing or explain better the rationale. Specifically refer to the following panels, where the consistent change in the results do not match the non-significance level measured: Figure S7c, Figure 3e, Figure S6e, Figure 4c.

We agree with the reviewer that the statistical testing may appear confusing. Therefore, we would like to clarify this point. Throughout the all study we consistently applied the following criteria: *Each data set was subjected to a normality test prior to further comparisons (Shapiro-Wilk normality test; alpha=0.05). For normally distributed data sets, when comparing two independent samples (i.e., data points*

from different subjects) two-tailed unpaired Student's t-test, whereas when comparing two dependent samples (i.e., data points from the same subject) two-tailed paired t-test was used. For non-normally distributed data sets, when comparing two independent samples (i.e., data points from different subjects) two-tailed Mann-Whitney test, whereas when comparing two dependent samples (i.e., data points from the same subject) two-tailed Wilcoxon matched-pairs signed rank test was used. This has been added to the sub-section "Data and statistical analysis" in the Materials and Methods (**Page 25, lines 668-676**).

Moreover, we wish to further clarify the panels pointed out by the reviewer:

Figure S7c (new Supplementary Figure 8c): This figure shows the percentage of GFP expressing cells in the DG of mice expressing either E-SARE-HA-LacZ or E-SARE-HA-Dnmt3a2 after CFC training. Each statistical comparison was performed between the LacZ group and Dnmt3a2 group for total dentate gyrus (DG), upper blade of the DG or lower blade of the DG (but not between different DG blades). For total DG comparisons, data was normally distributed (alpha = 0.05) by Shapiro- Wilk normality test, and since independent data points were being compared, unpaired t-test was applied. For upper blade of the DG comparisons, data from the LacZ group did not pass Shapiro- Wilk normality test, and since independent data points were being compared, Mann-Whitney test was applied. For lower blade of the DG comparisons, data was normally distributed (alpha = 0.05) by Shapiro- Wilk normality test, and since independent data points were being compared, unpaired t-test was applied.

Figure 3e: This figure shows the percentage of neurons that are activated by both CFC training and recall in the DG of the mice expressing either E-SARE-HA-LacZ or E-SARE-HA-Dnmt3a2. Also in this figure, each statistical comparison was performed between the LacZ group and Dnmt3a2 group for total DG, upper blade of the DG or lower blade of the DG (but not between different DG blades). For total DG and upper blade of the DG comparisons, data was normally distributed (alpha = 0.05) by Shapiro- Wilk normality test, and since independent data points were being compared, unpaired t-test was applied. For lower blade of the DG comparisons, both data points from the LacZ and Dnmt3a2 groups were not normally distributed by Shapiro- Wilk normality test, and since independent data points were being compared, Mann-Whitney test was applied. We agree with the reviewer that particularly in the lower blade of the DG, it may appear as if there is a strong trend for an increase in Dnmt3a2 group despite p value being 0.7203, but that is likely due to a single data point that increases the group average and does not reflect the rest of the group (see below). Yet, since there is no statistical or biological reason to identify that animal as an outlier, we believe it is appropriate to maintain this point in the data set.

Figure S6e (new Supplementary Figure 7e): This figure compares the rate of observed overlap versus the chance overlap within each experimental group, and then compares the fold above the chance overlap between Cre-loxP-GFP and Cre-loxP-HA-Dnmt3a2 groups. When comparing the statistical chance of overlap versus observed overlap per mouse within each experimental group, since data passed normality test by Shapiro- Wilk normality test ($\alpha = 0.05$) and since the data points being compared were collected from the same subject, paired t-test was applied. When comparing the observed/chance overlap ratios between the experimental groups, since the data showed a normal distribution by Shapiro- Wilk normality test ($\alpha = 0.05$), and since between subject comparisons were performed, unpaired t-test was applied.

Figure 4c: This figure compares the rate of observed overlap versus the chance overlap within each experimental group, and then compares the fold above the chance overlap between E-SARE-HA-LacZ and E-SARE-HA-Dnmt3a2 groups. When comparing the statistical chance of overlap versus observed overlap per mouse within each experimental group, since data points from the HA-LacZ group passed Shapiro-Wilk normality test ($\alpha = 0.05$), whereas HA-Dnmt3a2 group failed to pass Shapiro- Wilk normality test ($\alpha = 0.05$), paired t-test and Wilcoxon test was applied, respectively to compare data points of each subject (same-subject comparison). When comparing the observed/chance overlap ratios between the experimental groups, since the data showed a normal distribution by Shapiro- Wilk normality test ($\alpha = 0.05$), and since between subject comparisons were performed, unpaired t-test was applied.

4. The results suggest that over expression of Dnmt3a is a subset of engram cells effects memory and recall. However, the mechanisms through which the effect is mediated is not clear.

a. In the discussion you claim that it is: "likely by modulating the stability of the neuronal ensemble", but this claim is speculative and thus should be further explained or supported.

In order to further explain and support our hypothesis we restructured the discussion and discussed this view in light of our new data (DNA methylation analysis following the overexpression of Dnmt3a2 – see new **Figure 5** and response to point below) (**pages 14,15, lines 364-392**). We hope these changes clarified that the concept of increased stability of neuronal ensembles refers to the observed improved reactivation of the learning ensemble upon memory recall, which, as our data shows, is driven by the overexpression of Dnmt3a2 within neuronal ensembles. Moreover, we propose that a possible mechanism is the modulation of the strength of connectivity in members of the neuronal ensemble population. This is now further supported by our new experiments showing that Dnmt3a2 overexpression leads to DNA methylation changes in genes with established roles in synaptic plasticity. These included genes known to regulate synapse structure and activity (new **Figure 5 and Supplementary Tables 1, 2**).

b. Evidence for DNA methylation or gene expression changes following the over expression of Dnmt3a will strengthen the manuscript significantly (global measurements are preferred but specific measurements of genes/domains previously suggested to change with learning/activity will also be helpful, see for example the work of Guo et al. 2011).

We agree with the reviewer that evidence for DNA methylation or gene expression changes driven by the manipulation used in our study would significantly strengthen the manuscript. Therefore we performed a global assessment of DNA methylation changes using whole genome bisulfite sequencing in primary hippocampal cultures

that expressed E-SARE driven GFP or HA-Dnmt3a2 upon neuronal activity (new **Figure 5**). This experiment allowed us to identify in an unbiased manner the differentially methylated genomic regions and genes (DMRs and DMGs, respectively) in the condition of Dnmt3a2 overexpression versus the control condition (new **Supplementary Table 1**). Interestingly, we found a significant overrepresentation of DMGs involved in synaptic plasticity and memory mechanisms (new **Supplementary Table 2**). Hence, this new data suggests that epigenetic regulation of memory-related genes underlies the observed effects. Furthermore, we found that the DNA methylation changes predominantly occurred in intergenic and intronic regions. These findings are in line with two previous studies that showed similar enrichment for activity-dependent DNA methylation changes^{1,2}. Thus, confirming that the regulatory functions of DNA methylation in memory go beyond a direct regulation of promoter activity. Future studies will elucidate how the identified DNA methylation changes influence gene activity. We thank the reviewer for the suggestion as we agree that this experiment has significantly strengthened our study.

Although it would have been interesting to identify DNA methylation changes *in vivo*, that experiment would involve the selective analysis of DNA methylation in the very sparse dentate gyrus neuronal ensemble population. This would require extensive methodological establishment and characterization, in addition to the use of several additional animals. Therefore we think this is outside the scope of the current revision and will be more suitable for future studies.

5. The % of activated neurons carry significantly between experiments and experimental systems (e.g. %Fos+ 0.5% (Figure S6), %HAGFP 2% (Figure S4)), which is concerning regarding the robustness of the system and the labeling of the true engram cells. How do you reconcile this? is this technical or biological variation?

The identification of activated neurons is done throughout the study using either the expression of the activity-regulated proteins Fos or Arc or using the viral tools that are based on the promoters of the activity-regulated genes (E-SARE or pRAM promoters). Throughout the study we obtained similar percentages of Fos- or Arc-positive neurons within each behavioral procedure, which indicates low technical variability:

Novel environment exposure: 2% Arc-positive neurons (Supplementary Figure 2b) 1% Fos-positive neurons (Supplementary Figure 5b, previous S4) Contextual Fear Conditioning: 0.7% Fos-positive neurons (Supplementary Figure 7c, previous S6) 0.6% Fos-positive neurons (Supplementary Figure 8b)
--

Concerning the quantification of the activated neuronal population using the E-SARE or pRAM tools the variability was also low:

Novel environment exposure: 2% pRAM-GFP (Supplementary Figure 5b, previous S4) 5% E-SARE-GFP (Supplementary Figure 2b) Contextual Fear Conditioning: 4-6% pRAM-GFP (Supplementary Figure 8c) 4% pRAM-GFP (Supplementary Figure 6c) 6% E-SARE-HA-Dnmt3a2 (Supplementary Figure 6c) 5% E-SARE-HA-Dnmt3a2 (Supplementary Figure 2g)

Importantly, a comparison of the size of the neuronal population tagged with our tools to the one found in published studies that also focused in the dentate gyrus and used

contextual fear conditioning as the behavioral model, shows very similar values (Liu et al 2012: 6%³; Tayler et al 2013: 5%⁴; Sorensen et al 2016: 4%¹¹).

The size of the neuronal population labeled with Fos or the viral tools in contextual fear conditioning shows however a considerable difference. We think two factors contributed to this difference. First, the Fos-labeled population reflects the neurons activated by fear memory recall whereas the one labeled with E-SARE/pRAM represents the neurons activated by training. It has been shown before that contextual fear conditioning recall leads to lower number of Fos-positive neurons compared to training¹². Second, the E-SARE/pRAM-labeled neuronal population likely marks a slightly bigger population than the one marked by the expression of IEGs owing to the intrinsic leakiness of these systems. Despite the slight leakiness presented by the neuronal ensembles-labeling tools, we (**Supplementary Figures 2c-e and 5c**) and others^{3,6,11} have shown that they target a significant proportion of the neuronal population engaged in a particular behavioral task.

Taken together, the quantification of activated neurons throughout the current study was consistent within each labeling strategy and behavioral approach, indicating low technical variability.

6. Points need to be addressed in the text/discussion -

The following points have now been addressed in the Discussion:

a. Please address the fact that you over express the gene only in a sub-set of the engram cells.

We have now clearly stated in the Discussion that the overexpression of Dnmt3a2 occurs in approximately 75% of the engram cells (**lines 364, 365**).

b. Why did you choose to manipulate dnmt3a and not one of the other Dnmts or TET genes and how do you expect these other proteins to be involved.

To address this point we have now mentioned in the Discussion the various studies that demonstrated that among the various DNA methyltransferases, Dnmt3a is the one that has more consistently been shown to be required for hippocampal memory formation and synaptic plasticity^{10,13-16}. We further clarified that these studies together with our previous findings showing that Dnmt3a2 expression is regulated by neuronal activity¹⁶, lead us to select this Dnmt for the first manipulation of a DNA methylation regulator in neuronal ensembles (**lines 344-363**).

Additionally, we discuss that DNA methylation writing is likely not the only epigenetic mechanism that regulates neuronal ensemble stability, but it is plausible that an interplay between the enzymes responsible for DNA methylation writing and erasure and for modification of histones coordinates transcriptional changes in the neuronal population that holds the representation of a memory (**lines 436-442**).

c. Please explain your results in connection to previous contradictory results such as Morris et al. 2014 and Feng et al. 2010 each suggesting different roles for Dnmt3a/Dnmt1 in learning.

A discussion of the studies Morris et al 2014 and Feng et al 2010, as well as other studies that assessed the function of Dnmt3a in memory formation^{10,14,16} has now been added to the Discussion section (**lines 344-363**). The different studies that evaluated the function of Dnmt3a in memory formation, either by using conditional knockout mice in which the *Dnmt3a* gene is disrupted¹⁵ or by targeting the transcripts (*Dnmt3a1* or *Dnmt3a2*) using RNA interference^{14,16}, consistently revealed a requirement for these enzymes. The reason for the discrepancy between these

studies and Feng et al 2010 is not clear, but may be attributed to differences in the gender, age or genetic background of the mice.

Additional minor comments:

1. Supplementary Figure 1f: please also show the actual numbers of double and single positive counts and note how many sections were used.

The actual numbers have now been added, please see **Supplementary Figure 2d** and respective legend for information on numbers of slices.

2. Please make sure all abbreviations in figures are explained in figure legends (e.g. CFC).

We apologize for overseeing some of the abbreviations. This has now been checked and corrected.

3. Please add graphs showing quantification of behavior (such as % time freezing) displaying the actual data per mice (in addition to the average shown as bar plots).

We have now replaced all the graphs showing % of time freezing by graphs showing the individual data points and group average (**Figures 1 and 2**).

4. Figure S4d: columns are not labeled in a coherent way, specifically what's the difference between the first three and second three (on/off Dox?)?

We apologize for not providing a clear labeling of the lanes in the western blot. We have now labeled the first 3 lanes "off doxycycline" and the second 3 lanes "on doxycycline" (**Supplementary Figure 5d**).

5. Figure 5. the summary figure is very clear and helpful. But some fonts are too small, and the meaning of NE is unclear.

This has now been changed (**Figure 6**). We increased the font size and spelled out NE (neuronal ensembles).

6. Supplementary data tables, with the raw data from the behavior and image analysis would help improve the transparency and avoid confusion such as the one suggested above with the statistical testing.

We have now provided the source data file that contains all the raw data from the behavior and image analysis.

7. Please provide more details regarding the image analysis or the code used to quantify the proteins and their overlap.

We have now significantly expanded the sub-section "Microscopy and image analysis" in Materials and Methods (**lines 587-610**) in order to provide more detail regarding the analysis of images for co-localization analysis.

Reviewer #3 (Remarks to the Author):

The authors developed elegant viral manipulations to investigate the role of increased

expression of a DNA methyltransferase (Dnmt3a2) in engram cells after contextual fear conditioning leading to enhanced memory formation. Overall, they come to the conclusion that increased Dnmt3a2 expression in engram cells in dentate gyrus would not affect engram size but rather enhance engram stability assuring improved retrieval. In my opinion this work is of very high impact in the field due to the novel approach to manipulate engram cells at molecular level and because of the suggestion that memory enhancement might be due to increased engram stability. However, I think that the manuscript would benefit from some more control experiments as outlined below:

1) Fig. S1b shows that the Dnmt3a2 overexpression system is leaky in primary neurons. Such leakiness is a great concern, when claiming to study the effect of Dnmt3a2 in engram cells. I think that the authors should characterise the leakiness in vivo. Importantly, they should quantify the leaky expression in dentate gyrus and compare with expression levels of the intended leaky expression system in Figure 2.

We agree with the reviewer that analyzing the expression of HA-Dnmt3a2 *in vivo* is important. We have now quantified the expression of HA-Dnmt3a2 in the dentate gyrus in home cage conditions and after training in contextual fear conditioning (**Supplementary Figure 2f,g**). We found that in home cage conditions the expression is restricted to 3,7% of dentate gyrus granule cells. The expression of the endogenous activity-regulated protein Arc in the same conditions is confined to 1% of the cells. Although, this indeed indicates that E-SARE driven expression presents some leakiness, similar levels of basal expression were found in other studies that used neuronal ensemble labeling tools (3%^{3,4}). Importantly, random non-ensemble HA-Dnmt3a2 (in Figure 2) is confined to 5% of the neurons (note that this comprises a even slightly bigger population than the leaky E-SARE-driven expression (3,7%)) (**Supplementary Figure 4a, page 7 lines 170-173**). This confirms that the two models exhibit similar sparseness in Dnmt3a2 overexpression. Therefore, this indicates that the leakiness of the E-SARE system should not underlie the observed memory enhancement and improved reactivation.

2) Fig. S1c indicates that the viral transfection is not only in the dentate gyrus, but also other regions in the hippocampus, including area CA1. Therefore, the authors should analyse effects in area CA1 to assure that primarily dentate gyrus and not other hippocampal regions were manipulated.

This comment may have been triggered by the apparent positive t-dimer signal in *Stratum radiatum* and *Stratum oriens*. Imaging of uninfected brain slices shows that this is background rather than t-dimer signal:

Alternatively the comment may have been triggered by the presence of low GFP signal in some cells of the CA1 pyramidal layer in the KA (Kainic acid treated) condition. The presence of GFP-positive cells in the pyramidal layer of the CA1 in Supplementary Figure 2a is likely due to the unique conditions of this experiment i.e., strong unphysiological increases in neuronal activity induced by KA administration that lead to the transcription of otherwise neglectable copies of viral DNA. Moreover, every animal that underwent behavioral testing was checked for viral expression and targeting. Only mice in which viral expression was exclusively present in the dentate gyrus were included (This information has now been added to the Materials and Methods section (**page 19, lines 492,493**)).

3) It would be good to illustrate the kinetics of the inducibility of increased Dnmt3a2 expression after CFC to understand which consolidation window may be affected.

To address this point, we performed western blot and immunohistochemical analysis of dentate gyrus tissue from mice injected with E-SARE-HA-Dnmt3a2 that remained in home cage conditions or were trained in the contextual fear conditioning (**Supplementary Figure 2f,g**). For western blot analysis, the tissue was extracted 2, 6, 12 or 24 hours after fear conditioning training. We observed an increase in the levels of HA-Dnmt3a2 from two hours after learning. We have confirmed this result by immunohistochemical analysis; the number of HA-Dnmt3a2⁺ neurons was significantly higher in mice that underwent fear conditioning (analysis was performed 2 h after the training). This confirms that the overexpression of Dnmt3a2 is present from the early stages of consolidation.

We would like to thank the reviewers for their helpful comments and suggestions. We believe that we have addressed all points raised, which has significantly strengthened our manuscript. We hope that you will find our manuscript now suitable for publication in Nature Communications.

Best wishes,

Ana M.M. Oliveira

References

1. Guo, J.U., *et al.* Neuronal activity modifies the DNA methylation landscape in the adult brain. *Nature neuroscience* **14**, 1345-1351 (2011).
2. Halder, R., *et al.* DNA methylation changes in plasticity genes accompany the formation and maintenance of memory. *Nature neuroscience* **19**, 102-110 (2016).
3. Liu, X., *et al.* Optogenetic stimulation of a hippocampal engram activates fear memory recall. *Nature* **484**, 381-385 (2012).
4. Tayler, K.K., Tanaka, K.Z., Reijmers, L.G. & Wiltgen, B.J. Reactivation of neural ensembles during the retrieval of recent and remote memory. *Curr Biol* **23**, 99-106 (2013).
5. Han, J.H., *et al.* Neuronal competition and selection during memory formation. *Science* **316**, 457-460 (2007).
6. Kawashima, T., *et al.* Functional labeling of neurons and their projections using the synthetic activity-dependent promoter E-SARE. *Nature methods* **10**, 889-895 (2013).
7. Wang, X., *et al.* Immunofluorescently labeling glutamic acid decarboxylase 65 coupled with confocal imaging for identifying GABAergic somata in the rat dentate gyrus-A comparison with labeling glutamic acid decarboxylase 67. *Journal of chemical neuroanatomy* **61-62**, 51-63 (2014).
8. Vazdarjanova, A., *et al.* Spatial exploration induces ARC, a plasticity-related immediate-early gene, only in calcium/calmodulin-dependent protein kinase II-positive principal excitatory and inhibitory neurons of the rat forebrain. *The Journal of comparative neurology* **498**, 317-329 (2006).
9. Tamamaki, N., *et al.* Green fluorescent protein expression and colocalization with calretinin, parvalbumin, and somatostatin in the GAD67-GFP knock-in mouse. *The Journal of comparative neurology* **467**, 60-79 (2003).
10. Oliveira, A.M., Hemstedt, T.J., Freitag, H.E. & Bading, H. Dnmt3a2: a hub for enhancing cognitive functions. *Mol Psychiatry* **21**, 1130-1136 (2016).
11. Sorensen, A.T., *et al.* A robust activity marking system for exploring active neuronal ensembles. *eLife* **5**(2016).
12. Besnard, A., Laroche, S. & Caboche, J. Comparative dynamics of MAPK/ERK signalling components and immediate early genes in the hippocampus and amygdala following contextual fear conditioning and retrieval. *Brain Struct Funct* **219**, 415-430 (2014).
13. Feng, J., *et al.* Dnmt1 and Dnmt3a maintain DNA methylation and regulate synaptic function in adult forebrain neurons. *Nature neuroscience* **13**, 423-430 (2010).
14. Mitchnick, K.A., Creighton, S., O'Hara, M., Kalisch, B.E. & Winters, B.D. Differential contributions of de novo and maintenance DNA methyltransferases to object memory processing in the rat hippocampus and perirhinal cortex--a double dissociation. *Eur J Neurosci* **41**, 773-786 (2015).
15. Morris, M.J., Adachi, M., Na, E.S. & Monteggia, L.M. Selective role for DNMT3a in learning and memory. *Neurobiology of learning and memory* **115**, 30-37 (2014).
16. Oliveira, A.M., Hemstedt, T.J. & Bading, H. Rescue of aging-associated decline in Dnmt3a2 expression restores cognitive abilities. *Nature neuroscience* **15**, 1111-1113 (2012).

REVIEWERS' COMMENTS:

Reviewer #1 (Remarks to the Author):

I believe the authors have done a tremendous job in replying to my previous concerns. The new data strengthens the overall conclusions of the study and sharpens the interpretation of key findings significantly. The bisulfite sequencing addition was very important. It would be great to see a little more in the discussion about the overall approach and comparison to the CREB neuronal ensemble studies, especially as the non-ensemble over expression of Dnmt3a2 did not lead to memory enhancement or to neuronal ensemble allocation. That is an important point for the field and should be discussed more thoroughly in the discussion. Overall, terrific study.

Reviewer #2 (Remarks to the Author):

The revised manuscript is improved and the authors have addressed my comments.

Reviewer #3 (Remarks to the Author):

The authors have addressed my comments.

Reviewers' comments:

Reviewer #1 (Remarks to the Author):

I believe the authors have done a tremendous job in replying to my previous concerns. The new data strengthens the overall conclusions of the study and sharpens the interpretation of key findings significantly. The bisulfite sequencing addition was very important. It would be great to see a little more in the discussion about the overall approach and comparison to the CREB neuronal ensemble studies, especially as the non-ensemble over expression of Dnmt3a2 did not lead to memory enhancement or to neuronal ensemble allocation. That is an important point for the field and should be discussed more thoroughly in the discussion. Overall, terrific study.

We greatly appreciate the reviewer's comments and positive response and feedback. We agree that expanding the discussion to include a comparison between our study and the ones in which CREB levels were modulated in non-ensemble neurons is important given the similar approach but contrasting results. The following text has now been included in the Discussion:

"Interestingly, previous studies that also manipulated the levels of a key memory molecule in a non-ensemble population prior to learning produced distinct findings. In these studies, overexpression of cAMP response element binding protein (CREB) in a sparse subset of lateral amygdala neurons before fear conditioning biased the recruitment of these neurons to a memory trace and promoted enhanced fear memory^{1,2}. Later on, it was shown by the same group that ensemble allocation is determined by the relative neuronal excitability immediately before training, a property that is modulated by CREB³. These findings together with our data, suggest distinct functions for CREB and Dnmt3a2 namely the ability to modulate neuronal properties in the absence of salient environmental stimuli."

Reviewer #2 (Remarks to the Author):

The revised manuscript is improved and the authors have addressed my comments.

We thank the reviewer for the positive response.

Reviewer #3 (Remarks to the Author):

The authors have addressed my comments.

We thank the reviewer for the positive response.

1 Han, J. H. *et al.* Neuronal competition and selection during memory formation. *Science* **316**, 457-460, doi:10.1126/science.1139438 (2007).

2 Han, J. H. *et al.* Selective erasure of a fear memory. *Science* **323**, 1492-1496, doi:10.1126/science.1164139 (2009).

3 Yiu, A. P. *et al.* Neurons are recruited to a memory trace based on relative neuronal excitability immediately before training. *Neuron* **83**, 722-735, doi:10.1016/j.neuron.2014.07.017 (2014).